# DiffThinker: Towards Generative Multimodal Reasoning with Diffusion Models

**Zefeng He** [1 2]   **Xiaoye Qu** [1 †]   **Yafu Li** [1 3]   **Tong Zhu** [1]   **Qipeng Guo** [1]   **Muxin Fu** [1 4]   **Siyuan Huang** [1 5]   **Yu Cheng** [3 †]

**Project Page:** https://diffthinker-project.github.io

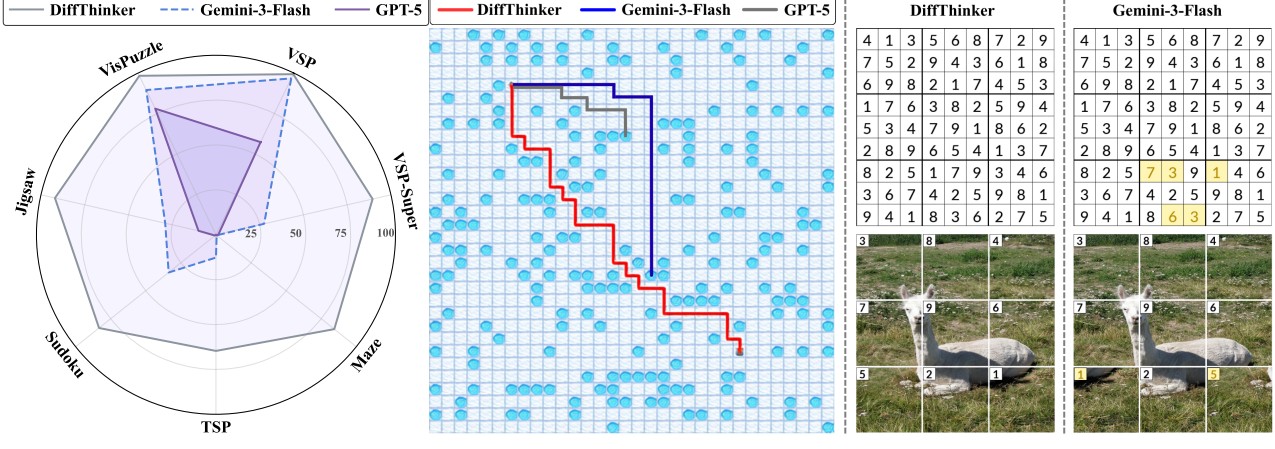

(a) Overall performance.     (b) Visualizations on VSP-Super, Sudoku and Jigsaw.

*Figure 1.* (a) Quantitative results across seven tasks. (b) DiffThinker produces solution images directly, whereas baseline results are post-processed visualizations of textual outputs with errors highlighted. By reformulating reasoning as a native image-to-image generative process, DiffThinker achieves superior logical consistency and spatial precision in complex long-horizon, vision-centric reasoning tasks.

## Abstract

While recent Multimodal Large Language Models (MLLMs) have attained significant strides in multimodal reasoning, their reasoning processes remain predominantly text-centric and fail to visualize and track intermediate visual states during the reasoning process, leading to suboptimal performance in complex long-horizon, vision-centric tasks. Moving beyond the constraints of text-centric reasoning, we establish Generative Multimodal Reasoning as a novel paradigm and introduce DiffThinker, a diffusion-based reasoning framework. Conceptually, DiffThinker reformulates multimodal reasoning as a native generative image-to-image task, where the iterative denoising trajectory naturally serves as a visual reasoning path. This enables the model to track the evolution of visual information throughout the reasoning process. We perform a systematic comparison between DiffThinker and MLLMs, providing the first in-depth investigation into the intrinsic characteristics of this paradigm, revealing four core properties: efficiency, controllability, native parallelism, and collaboration. Extensive experiments across seven tasks demonstrate that DiffThinker significantly outperforms leading closed-source models, including GPT-5 (+314.2%) and Gemini-3-Flash (+111.6%), as well as the fine-tuned Qwen3-VL-32B baseline (+39.0%), highlighting Generative Multimodal Reasoning as a promising approach for vision-centric reasoning.

[1]Shanghai AI Laboratory [2]Nanjing University [3]The Chinese University of Hong Kong [4]Tongji University [5]Shanghai Jiao Tong University. Correspondence to: Xiaoye Qu <quxiaoye@pjlab.org.cn>, Yu Cheng <chengyu@cse.cuhk.edu.hk>.

*Proceedings of the 43rd International Conference on Machine Learning*, Seoul, South Korea. PMLR 306, 2026. Copyright 2026 by the author(s).

## 1. Introduction

In recent years, Multimodal Large Language Models (MLLMs) (Google, 2025a; OpenAI, 2025a; Bai et al., 2025; Comanici et al., 2025) have achieved remarkable progress in multimodal understanding. Furthermore, the introduction of Chain-of-Thought (CoT) empowers these models with com-

plex reasoning capabilities. More recently, Reinforcement Learning with Verifiable Reward (Shao et al., 2024; Guo et al., 2025a; Zhang et al., 2025c) has substantially enhanced the reasoning capabilities of MLLMs (Wang et al., 2025a). Building upon these foundations, the emerging paradigm of "Thinking with Image" (OpenAI, 2025b; Zheng et al., 2025; Wang et al., 2025b; Su et al., 2025b) enables MLLMs to interact with multimodal inputs iteratively, further pushing the boundaries of multimodal reasoning.

Despite these advances, current multimodal reasoning remains primarily text-centric and fails to visualize and track intermediate visual states during the reasoning process, posing significant challenges for complex long-horizon, vision-centric tasks (Wu et al., 2024; Ivanitskiy et al., 2023). Furthermore, current MLLMs primarily rely on lengthy CoT for reasoning, resulting in uncontrollable generation and prohibitive latency (Sui et al., 2025; Qu et al., 2025). This inefficiency is further intensified by the multi-turn interactions inherent in the "Thinking with Image" paradigm.

Moving beyond the constraints of text-centric reasoning, we introduce **DiffThinker**, and establish **Generative Multimodal Reasoning** as a novel paradigm that shifts the reasoning from symbolic space to visual space. Unlike MLLMs that typically conceptualize reasoning as a multimodal-to-text mapping, we propose to model multimodal reasoning directly as a generative image-to-image task with diffusion models. In this framework, the iterative denoising trajectory of the diffusion model naturally serves as a native visual reasoning path. This enables the continuous tracking of state transitions directly within the visual space and overcomes the inherent challenges in long-horizon, vision-centric tasks.

We conduct a systematic comparison between DiffThinker and MLLMs across a diverse set of challenging tasks, and provide the first in-depth investigation into the intrinsic characteristics of Generative Multimodal Reasoning, revealing four core properties: ① **Efficient Reasoning:** It demonstrates superior efficiency in both training and inference, as well as higher accuracy, significantly outperforming RL-based MLLMs. ② **Controllable Reasoning:** It provides stable and controllable inference costs, contrasting with the variable length CoT in MLLMs. ③ **Native Parallel Reasoning:** It inherently explores multiple candidate solutions in parallel, progressively pruning invalid paths. ④ **Collaborative Reasoning:** It can act as a partner with MLLMs, yielding results superior to either model alone.

To comprehensively evaluate the performance of DiffThinker, we conduct experiments across seven tasks in four domains including sequential planning, combinatorial optimization, constraint satisfaction, and spatial configuration. The results demonstrate that DiffThinker significantly outperforms state-of-the-art MLLMs, including GPT-5 (**+314.2%**), Gemini-3-Flash (**+111.6%**), and the Qwen3-VL-32B baseline (**+39.0%**) fine-tuned on identical datasets, which underscore the efficacy of Generative Multimodal Reasoning in vision-centric tasks.

In summary, our contributions are threefold:

- We propose DiffThinker and establish Generative Multimodal Reasoning as a novel paradigm, reformulating multimodal reasoning from text-centric symbolic mapping to a native image-to-image generative process.

- We perform a systematic comparison between DiffThinker and MLLMs and conduct the first investigation into the intrinsic attributes of Generative Multimodal Reasoning, revealing four core properties: efficiency, controllability, native parallelism, and collaboration.

- Extensive experiments on seven tasks demonstrate DiffThinker significantly outperforms SOTA MLLMs like GPT-5 (+314.2%) and Gemini-3-Flash (+111.6%), highlighting Generative Multimodal Reasoning as a promising approach for vision-centric reasoning.

**Conflict of Interest Disclosure.** The authors declare no financial or non-financial conflicts of interest related to this work. The models evaluated in this paper are not developed by any institution affiliated with the authors. All evaluations were conducted through publicly available APIs or released model weights under their respective terms of use.

## 2. Related Works

### 2.1. Multimodal Reasoning

Reinforcement Learning with Verifiable Reward (Guo et al., 2025a; Shao et al., 2024) has significantly enhanced LLM reasoning, and is rapidly extending to MLLMs (Huang et al., 2025b; Shen et al., 2025; Liu et al., 2025b; Huang et al., 2025a; 2026; He et al., 2025b; Wang et al., 2025a). However, existing paradigms remain predominantly text-centric, which hinders performance in vision-centric tasks. Advancing this frontier, the paradigm of "Thinking with Image" (OpenAI, 2025b) introduces a mechanism for models to engage in multi-turn visual interactions during the reasoning process. While earlier approaches (Su et al., 2025a; Zheng et al., 2025; Wang et al., 2025b; Hong et al., 2025; Zhang et al., 2025d; Lai et al., 2025) relied on tool calls or code execution for image manipulation, recent works (Yang et al., 2025b; Xu et al., 2025; Du et al., 2025; Zhang et al., 2025b; Wang et al., 2025c; Chen et al., 2025a; Qin et al., 2025; Gu et al., 2025) have shifted toward generating native images or latent visual tokens. Nevertheless, the underlying architectures of these methods remain rooted in autoregressive MLLMs, leading to limited performance in complex long-horizon, vision-centric tasks.

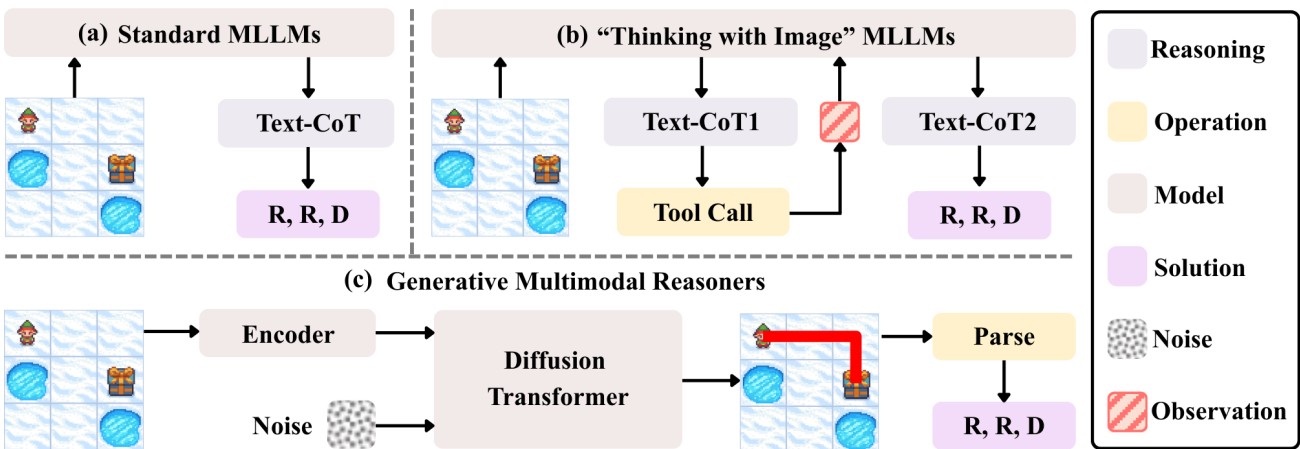

*Figure 2.* Overview of different multimodal reasoning paradigms. (a) Standard MLLMs map inputs directly to symbolic solutions. (e.g., 'R' and 'D' representing 'Right' and 'Down' actions) (b) "Thinking with Images" MLLMs interact with multimodal inputs through iterative tool calls. (c) DiffThinker reformulates multimodal reasoning as a direct generative image-to-image task, where solutions are produced in visual space and then parsed to symbolic solutions to ensure a fair comparison.

Building upon the success of "Thinking with Image," the "Thinking with Video" paradigm enhances reasoning by enabling models to interact with video content through multi-turn tool invocation (Zhang et al., 2025a; He et al., 2025a; Yan et al., 2025; Xie et al., 2025). This concept has recently advanced to performing multimodal reasoning directly through video generation (Wiedemer et al., 2025; Tong et al., 2025; Yang et al., 2025a; Luo et al., 2025; Liu et al., 2025a; Guo et al., 2025b; Wu et al., 2025b; Chen et al., 2025b). However, these studies predominantly focus on benchmarking closed-source models (Google, 2025b; OpenAI, 2025c) with undisclosed internal mechanisms. Furthermore, video generation itself entails prohibitive computational costs. Diverging from this, DiffThinker establishes image generation as a more efficient paradigm.

## 2.2. Diffusion Models

Diffusion models have emerged as the dominant framework for generative modeling. Early research (Sohl-Dickstein et al., 2015; Song & Ermon, 2019; Ho et al., 2020; Song et al., 2020; Ho & Salimans, 2022) laid the theoretical foundations of this paradigm. Subsequently, flow-based methodologies (Lipman et al., 2022; Liu et al., 2022; Albergo & Vanden-Eijnden, 2022) have further advanced the field. The integration of latent diffusion models (Rombach et al., 2022), diffusion transformers (Peebles & Xie, 2023), and multimodal diffusion transformers (Esser et al., 2024) has established the current mainstream for generative modeling, paving the way for diverse downstream applications.

While one prominent direction of research concentrates on high-fidelity image (Rombach et al., 2022; Ramesh et al., 2022; Saharia et al., 2022; He et al., 2026) and video (Ho et al., 2022; Wan et al., 2025; Brooks et al., 2024) genera-

tion, other applications extend to specialized tasks (Avdeyev et al., 2023; Ubukata et al., 2024; Pogodzinski et al., 2025; Graikos et al., 2022; Li et al., 2024; 2023), such as Sudoku (Wewer et al., 2025), geometry (Goren et al., 2025), and the Traveling Salesperson Problem (Sun & Yang, 2023). Unlike specialized methods requiring custom architectures and training from scratch, DiffThinker is independent of specific architectures and adapts to diverse tasks through rapid fine-tuning. Furthermore, we introduce Generative Multimodal Reasoning, focusing on various paradigms within multimodal reasoning and performing a comprehensive comparison between different reasoning frameworks.

## 3. Generative Multimodal Reasoning

### 3.1. Problem Reformulation

In this work, we introduce DiffThinker, a generative multimodal reasoner that innovatively reformulates multimodal reasoning as an image-to-image task, as illustrated in Figure 2. To clarify the paradigm shift, we define and formalize three distinct reasoning paradigms.

**Standard MLLMs: Image-to-Text.** Standard MLLMs model the reasoning process as a sequential mapping in the symbolic space. Given a visual input $x \in \mathcal{X}$ and a textual instruction $c \in \mathcal{T}$, the process is defined as:

$$f_{\text{Std}}(x, c) \to z \to y, \tag{1}$$

where $z$ represents the textual reasoning trace (e.g., CoT) and $y \in \mathcal{Y}$ is the final solution. Despite their success, the reasoning process remains text-centric, often leading to suboptimal performance in vision-centric tasks.

**"Thinking with Image" MLLMs: Iterative Interaction.** This paradigm enhances reasoning by enabling models to

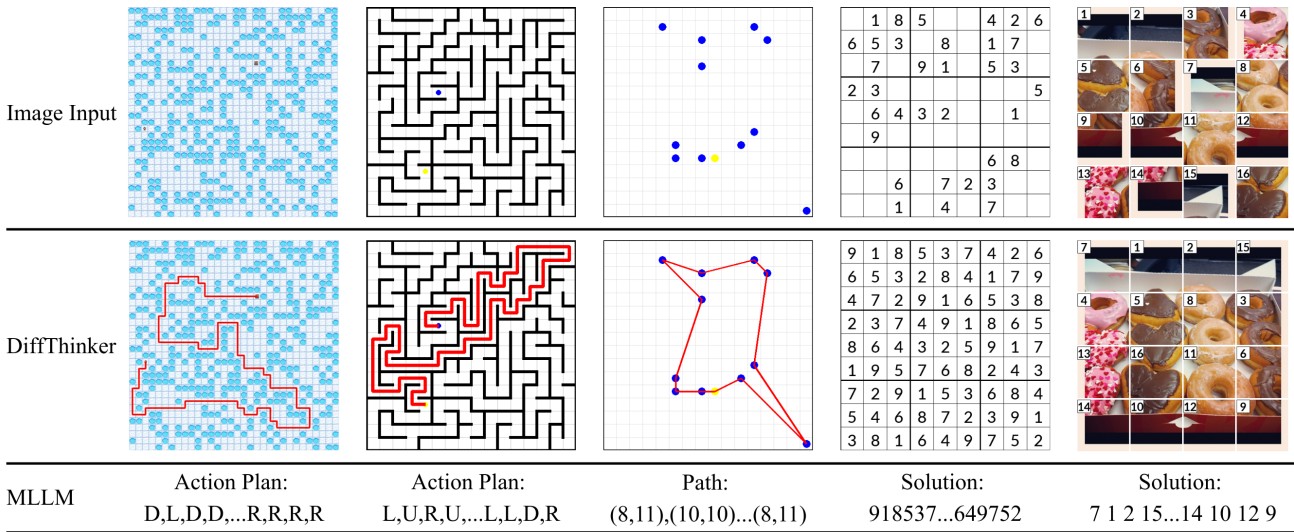

| | | | | |
|---|---|---|---|---|
| Image Input | | | | |
| DiffThinker | | | | |
| MLLM | Action Plan:
D,L,D,D,...R,R,R,R | Action Plan:
L,U,R,U,...L,L,D,R | Path:
(8,11),(10,10)...(8,11) | Solution:
918537...649752 |

Solution:
7 1 2 15...14 10 12 9

*Figure 3.* **Main tasks.** Each column represents a specific task. The first row displays the image input. The second row shows the results generated by DiffThinker. The third row presents the outputs from the MLLM baseline.

interact with multimodal inputs to generate intermediate results via tool calls. The process is formulated as an interleaved sequence of reasoning, tool call, and observation:

$$f_{\text{TwI}}(x, c) \rightarrow \{(z_1, t_1, o_1), \dots, (z_n, t_n, o_n)\} \rightarrow y, \quad (2)$$

where $z_i$ denotes the $i$-th reasoning step, $t_i$ represents the tool call, $o_i$ is the corresponding intermediate visual observation, and $y \in \mathcal{Y}$ signifies the final solution. Although this paradigm incorporates essential visual feedback, its inherent reliance on iterative multi-turn loops and the associated computational overhead pose significant challenges for scaling to complex long-horizon, vision-centric tasks.

**DiffThinker: Image-to-Image.** Unlike MLLMs, which primarily reason within symbolic space, DiffThinker shifts reasoning into visual space via an iterative generative process with diffusion models. Conditioned on the multimodal input $(x, c)$, the model serves as a generator $G$ and produces a sampling trajectory that refines an initial state (typically noise) into a solution image $x_{sol}$:

$$G(x, c) \rightarrow \{x_t\}_{t=T}^{0}, \quad \text{with } x_0 = x_{sol} \in \mathcal{X}. \quad (3)$$

Crucially, the denoising trajectory $\{x_t\}$ naturally serves as a visual reasoning path, where each $x_t$ captures an intermediate state of the solution image. To facilitate comparison with the symbolic ground-truth, we introduce a parsing function $\Psi : \mathcal{X} \rightarrow \mathcal{Y}$ to map the solution image to symbolic space:

$$y_{parsed} = \Psi(x_{sol}). \quad (4)$$

Rather than relying on MLLMs to judge whether a solution image conforms to the textual ground-truth, our parsing mechanism ensures a fair comparison across different paradigms and precludes potential answer leakage.

### 3.2. Flow Matching

DiffThinker is implemented based on Qwen-Image-Edit (Wu et al., 2025a). Mathematically, it employs Flow Matching (Lipman et al., 2022; Liu et al., 2022; Albergo & Vanden-Eijnden, 2022) as the theoretical framework to approximate the velocity field that transforms noise into the data distribution, ensuring stable learning dynamics via Ordinary Differential Equations (ODEs). Architecturally, the model leverages a Multimodal Diffusion Transformer (MMDiT) (Esser et al., 2024) to capture intricate cross-modal dependencies. For efficiency, these generative processes are performed within the latent space of a Variational Autoencoder (VAE) (Kingma & Welling, 2013).

**Training.** Formally, let $y$ denote the ground-truth image. The data latent $x_0$ is obtained by encoding $y$ through the VAE encoder $\mathcal{E}$, i.e., $x_0 = \mathcal{E}(y)$. A random noise vector $x_1$ is sampled from the standard multivariate normal distribution, $x_1 \sim \mathcal{N}(\mathbf{0}, \mathbf{I})$. To incorporate multimodal task constraints, the conditioning latent $h$ is derived from the MLLM $\phi$ given the user instruction $S$ (comprising text and visual inputs), such that $h = \phi(S)$. During training, a timestep $t$ is sampled from a logit-normal distribution with $t \in [0, 1]$. The intermediate latent variable $x_t$ is constructed via linear interpolation between the data $x_0$ and noise $x_1$:

$$x_t = tx_0 + (1 - t)x_1. \quad (5)$$

Consequently, the target velocity field $v_t$ driving the flow from noise to data is defined as:

$$v_t = \frac{dx_t}{dt} = x_0 - x_1. \quad (6)$$

The MMDiT-based vector field $v_\theta$ is trained to predict this target velocity $v_t$. The training objective is formulated as

*Table 1.* **Comprehensive Results across All Tasks.** We evaluate models across a total of four domains including Sequential Planning (VSP, VSP-Super, and Maze), Combinatorial Optimization (TSP), Constraint Satisfaction (Sudoku), and Spatial Configuration (Jigsaw and VisPuzzle). Evaluation is conducted on varying difficulty levels, defined by grid size for Sequential Planning and Jigsaw, number of cities for TSP, and number of given clues for Sudoku. "N/A" denotes vanilla models without training. Best results are highlighted in bold. The Avg column represents the grand mean calculated from individual task averages.

| Model | Setting | VSP | | | | | | VSP-Super | | Maze | | | TSP | | | Sudoku | | | Jigsaw | | | VisP. | Avg |
|---|---|---|---|---|---|---|---|---|---|---|---|---|---|---|---|---|---|---|---|---|---|---|---|
| | | 3 | 4 | 5 | 6 | 7 | 8 | 16 | 32 | 8 | 16 | 32 | 12 | 15 | 18 | 45 | 40 | 35 | 2 | 3 | 4 | | |
| *Closed-Source MLLMs* | | | | | | | | | | | | | | | | | | | | | | | |
| Gemini-3-Flash | N/A | **100** | **100** | **100** | 99 | 83 | 98 | 52 | 3 | 0 | 0 | 0 | 25 | 9 | 4 | 69 | 29 | 3 | 71 | 16 | 0 | 89.5 | 41.3 |
| GPT-5 | N/A | 99 | 70 | 67 | 43 | 36 | 29 | 3 | 0 | 2 | 0 | 0 | 0 | 0 | 0 | 2 | 0 | 0 | 30 | 0 | 0 | 78.0 | 21.1 |
| *Open-Source MLLMs* | | | | | | | | | | | | | | | | | | | | | | | |
| Qwen3-VL-8B | N/A | 64 | 46 | 33 | 21 | 12 | 21 | 1 | 0 | 0 | 0 | 0 | 0 | 0 | 0 | 0 | 0 | 0 | 7 | 0 | 0 | 28.0 | 9.1 |
| | SFT | 99 | 96 | 98 | 96 | 92 | 86 | 61 | 8 | 53 | 37 | 0 | 59 | 60 | 43 | 30 | 17 | 2 | 95 | 56 | 9 | 78.8 | 51.6 |
| | GRPO | 91 | 70 | 70 | 31 | 34 | 24 | 0 | 0 | 0 | 0 | 0 | 0 | 0 | 0 | 0 | 0 | 0 | 6 | 0 | 0 | 28.0 | 11.9 |
| | SFT + GRPO | 100 | 98 | 98 | 97 | 93 | 89 | 32 | 6 | 48 | 26 | 0 | 55 | 46 | 34 | 24 | 16 | 4 | 95 | 55 | 6 | 75.0 | 46.6 |
| Qwen3-VL-32B | N/A | 75 | 51 | 47 | 25 | 23 | 26 | 0 | 0 | 0 | 0 | 0 | 0 | 0 | 0 | 0 | 0 | 0 | 9 | 0 | 0 | 29.5 | 10.5 |
| | SFT | 96 | 99 | 98 | **100** | 99 | 90 | 85 | 21 | 91 | 57 | 3 | 69 | 59 | 52 | 32 | 22 | 2 | 97 | 72 | 28 | 95.8 | 62.9 |
| | GRPO | 99 | 90 | 95 | 69 | 73 | 58 | 1 | 0 | 0 | 0 | 0 | 0 | 0 | 0 | 3 | 1 | 0 | 64 | 4 | 0 | 83.0 | 26.9 |
| | SFT + GRPO | 99 | 94 | 95 | 94 | 90 | 90 | 15 | 2 | 59 | 32 | 0 | 54 | 39 | 33 | 21 | 15 | 0 | 92 | 51 | 10 | 94.5 | 47.4 |
| *Generative Multimodal Reasoners* | | | | | | | | | | | | | | | | | | | | | | | |
| Nano Banana 2 | N/A | 75 | 66 | 41 | 42 | 35 | 35 | 6 | 7 | 1 | 0 | 0 | 0 | 0 | 0 | 10 | 5 | 0 | 11 | 0 | 0 | 39 | 14.8 |
| LongCat-Image-Edit | N/A | 46 | 39 | 31 | 14 | 23 | 16 | 3 | 1 | 0 | 1 | 0 | 0 | 0 | 0 | 0 | 0 | 0 | 3 | 0 | 0 | 9.5 | 5.8 |
| **DiffThinker-Mini (Ours)** | Flow Matching | 99 | 97 | 99 | 94 | 95 | 99 | 77 | 63 | 44 | 28 | 16 | 36 | 26 | 21 | 70 | 39 | 9 | 68 | 28 | 2 | 81.3 | 53.9 |
| Qwen-Image-Edit-2509 | N/A | 33 | 36 | 22 | 12 | 11 | 7 | 0 | 0 | 0 | 0 | 0 | 0 | 0 | 0 | 0 | 0 | 0 | 0 | 0 | 0 | 7.5 | 4.0 |
| **DiffThinker (Ours)** | Flow Matching | 99 | **100** | 98 | 99 | **100** | **100** | 96 | 83 | **100** | 97 | 56 | 74 | 62 | 58 | **98** | **95** | **57** | 99 | 97 | 80 | 98.3 | 87.4 |
| Qwen-Image-Edit-2511 | N/A | 50 | 55 | 44 | 16 | 23 | 23 | 0 | 0 | 0 | 0 | 0 | 0 | 0 | 0 | 0 | 0 | 0 | 0 | 0 | 0 | 11.5 | 6.7 |
| **DiffThinker++ (Ours)** | Flow Matching | **100** | **100** | **100** | 98 | **100** | **100** | **99** | 80 | **100** | **100** | 65 | 76 | 72 | 59 | 97 | 94 | 55 | **99** | **98** | **80** | **98.8** | **88.5** |

the mean squared error (MSE):

$$\mathcal{L}_{FM} = \mathbb{E}_{t,x_0,x_1} \left[ \|v_\theta(x_t,t,h) - (x_0 - x_1)\|^2 \right]. \quad (7)$$

**Inference.** During inference, DiffThinker performs reasoning by solving the ODE defined by the learned velocity field $dx_t = v_\theta(x_t,t,h)dt$. From initial noise $x_{t=0} = x_1$, the model numerically integrates the flow to recover the solution latent $x_{t=1} \approx x_0$. Implementing a first-order Euler solver with a step size $\Delta t = 1/T$, the update rule is:

$$x_{t+\Delta t} = x_t + \Delta t \cdot v_\theta(x_t,t,h). \quad (8)$$

After $T$ steps, the final latent $x_{t=1}$ (which approximates the data distribution) is decoded back to pixel space via the VAE decoder to yield the visual solution: $y_{sol} = \mathcal{D}(x_{t=1})$.

### 3.3. Task Formulation

To systematically verify the efficacy of DiffThinker within the proposed generative reasoning paradigm, we select tasks based on three perspectives. First, we target complex long-horizon, vision-centric tasks that rely on visual perception and imagination. Second, we prioritize tasks offering controllable and scalable difficulty levels, which facilitates a precise exploration of the model's capability boundaries. Third, we specifically select tasks featuring high structural parseability, such as grid-based configurations. Since the evaluation of DiffThinker involves parsing generated visual solutions into symbolic formats, this criterion ensures an objective assessment against ground-truth labels. Accordingly, our tasks contain five distinct classes as detailed below.

**Visual Spatial Planning (VSP)** (Wu et al., 2024) **and VSP-Super.** VSP evaluates perception and reasoning capabilities in spatial planning scenarios. We focus on its FrozenLake subset due to its parseability. Moreover, we introduce VSP-Super, which expands the environment scale. As illustrated in the first column of Figure 3, the model must navigate a grid-based frozen lake while avoiding holes. DiffThinker generates a continuous visual trajectory rendered as a red line. Conversely, MLLMs produce text-based action plans. We formalize these challenges as sequential planning tasks.

**Maze** (Ivanitskiy et al., 2023)**.** This task involves longer routes than VSP, increasing navigation complexity. As illustrated in the second column of Figure 3, the model must identify a path avoiding walls between cells. DiffThinker renders a trajectory from the yellow start to the blue target. Conversely, MLLMs output an action plan via a series of text tokens. We categorize this as a sequential planning task.

**Traveling Salesperson Problem (TSP)** (Jünger et al., 1995)**.** This task requires solving the Traveling Salesperson Problem on a 2D plane, aiming to identify the shortest path visiting every city. As depicted in the third column of Figure 3, the problem is visualized by a yellow start dot and blue city dots. DiffThinker generates a geometric path connecting all nodes into a closed loop. In contrast, MLLMs provide numerical coordinates to represent the order. This is classified as a combinatorial optimization problem.

**Sudoku.** In this task, the model must fill in missing digits while adhering to Sudoku constraints. As shown in the fourth column of Figure 3, DiffThinker generates a

| Input | Step 1 | Step 4 | Step 7 | Output |
|---|---|---|---|---|

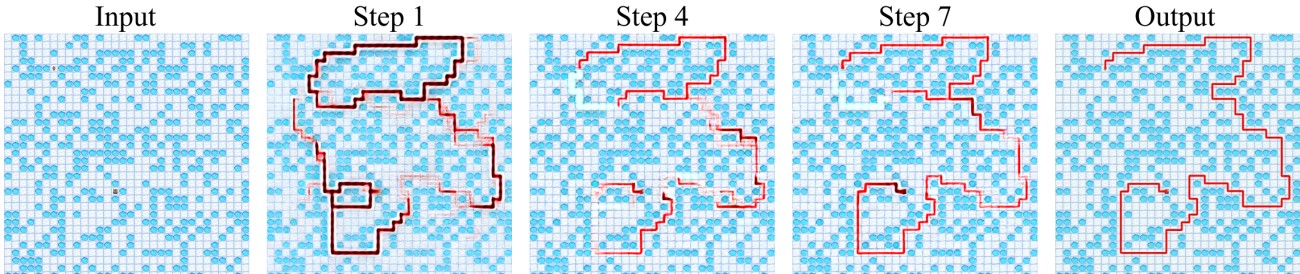

*Figure 4.* **Visualization of the native parallel reasoning process in DiffThinker.** The model explores multiple candidate paths simultaneously in the early stages and iteratively refines them into a single valid trajectory.

completed grid with all empty cells populated. Conversely, MLLMs provide a text-based numerical sequence. This challenge is classified as a constraint satisfaction task.

**Jigsaw and VisPuzzle (Gu et al., 2025).** The Jigsaw task centers on spatial configuration and visual perception. As illustrated in the final column of Figure 3, the input consists of shuffled patches, each numerically labeled to facilitate automated parsing. DiffThinker reconstructs these patches into a globally consistent image. In contrast, MLLMs produce a sequence of indices representing the restoration order. We also introduce VisPuzzle (Gu et al., 2025), which serves as a simplified benchmark for puzzle reconstruction. These challenges are categorized as spatial configuration tasks.

## 4. Experiments

**Experimental Setup.** DiffThinker is built upon Qwen-Image-Edit-2509 (Wu et al., 2025a), utilizing a 20B MMDiT (Esser et al., 2024). Additionally, we implement DiffThinker-Mini based on LongCat-Image-Edit (Team et al., 2025) (featuring a 6B backbone) and DiffThinker++ based on the updated Qwen-Image-Edit-2511 for main results (Table 1), whereas all subsequent analysis and ablation studies are conducted using DiffThinker. Following previous works (Xu et al., 2025; Li et al., 2025; Yang et al., 2025a), we train independent models for VSP/VSP-Super, Maze, TSP, Sudoku, and Jigsaw, respectively. We also fine-tune Qwen3-VL baselines on identical datasets for a direct comparison. VisPuzzle serves as an out-of-distribution task for puzzle reconstruction. Evaluation is conducted on varying difficulty levels, as shown in Table 1. More implementation details are provided in Appendix B.

### 4.1. Main Results

**DiffThinker as an Extraordinary Multimodal Reasoner.** As illustrated in Table 1, DiffThinker achieves state-of-the-art performance across seven challenging tasks in four domains. Specifically, our DiffThinker drastically surpasses GPT-5 (+314.2%), Gemini-3-Flash (+111.6%), and the fine-tuned Qwen3-VL-32B (+39.0%) with fewer parameters.

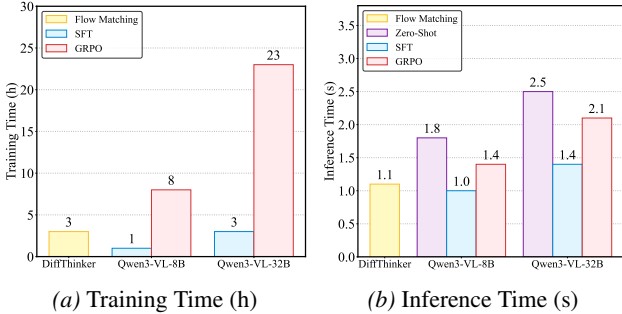

*(a)* Training Time (h)     *(b)* Inference Time (s)

*Figure 5.* **Computational efficiency analysis**. (a) Training duration: DiffThinker significantly outpaces RL-based MLLMs in efficiency. (b) Inference latency: DiffThinker achieves lower execution time than the Qwen3-VL-32B baseline.

Across all evaluated domains, DiffThinker demonstrates a clear advantage over traditional MLLMs. In sequential planning tasks such as VSP, VSP-Super, and Maze, the performance of MLLMs decays rapidly as task complexity scales, whereas DiffThinker maintains high accuracy through generative reasoning. In spatial configuration tasks including Jigsaw and VisPuzzle, the model achieves near-perfect performance, while similarly delivering exceptional results in combinatorial optimization (TSP) and constraint satisfaction (Sudoku). These results underscore that our generative multimodal reasoning paradigm provides a more robust foundation for multimodal reasoning than that of traditional MLLMs in long-horizon, vision-centric tasks. More comparisons with "Thinking with Image" MLLMs and video generation models are provided in Appendix A.

### 4.2. Discussion and Observation

**DiffThinker as an Efficient Reasoner.** To quantitatively assess the computational overhead of DiffThinker relative to standard MLLMs, we conduct experiments to measure both training and inference durations on a cluster of eight NVIDIA H200 GPUs. We report training durations of VSP/VSP-Super, and the average inference latency of VSP-Super level-16 per reasoning instance.

As illustrated in Figure 5(a), DiffThinker maintains a highly

competitive training efficiency. Its training duration is nearly identical to Qwen3-VL-32B (SFT) baseline and is substantially lower than the overhead of GRPO (Shao et al., 2024), a reinforcement learning paradigm currently widely adopted for multimodal reasoning. Regarding inference speed, as illustrated in Figure 5(b), DiffThinker exhibits a highly competitive latency of 1.1s, which is comparable to Qwen3-VL-8B (SFT) baseline (1.0s) and faster than Qwen3-VL-32B (SFT) model (1.4s). This result underscores the inherent inference efficiency of our generative reasoning paradigm.

**DiffThinker as a Controllable Reasoner.** DiffThinker establishes a controllable reasoning paradigm by reformulating tasks into a fixed-step generative process. By employing an Euler solver with a predefined number of steps, the model ensures a deterministic computational budget which is invariant to the task's logical complexity. In contrast, MLLMs are plagued by unpredictable inference durations. Their autoregressive nature often leads to fluctuating latency caused by verbose CoT or repetitive output collapse, resulting in significantly longer average inference times, as shown in Figure 5(b). Moreover, unlike MLLMs where imposing token limits risks premature truncation, the controllable generation of DiffThinker guarantees both execution stability and the derivation of reliable solutions.

**DiffThinker as a Native Parallel Reasoner.** Unlike MLLMs, which execute reasoning sequentially and often require explicit reflection or backtracking to rectify early errors (Jaech et al., 2024; Guo et al., 2025a), DiffThinker possesses an inherent capacity for native parallel reasoning. To visualize the progressive reasoning process, we estimate the predicted original sample at each intermediate timestep by projecting the current state back to the data manifold, and then decoding it into pixel space. As illustrated in Figure 4, during the initial reasoning stages (e.g., Step 1), DiffThinker avoids premature commitment to a single path, instead exploring multiple candidate trajectories across the grid in parallel. Through successive iterations, the model simultaneously evaluates global constraints and environmental obstacles to prune invalid routes, progressively consolidating its focus onto the most plausible path and eventually converging to an optimal solution.

**DiffThinker as a Collaborative Partner.** Beyond a direct comparison, we explore the synergy between DiffThinker and MLLMs in solving complex tasks. As illustrated in Figure 6(a), DiffThinker first produces multiple candidate solution images, which the MLLM then evaluates against the original problem constraints to make a final decision.

We benchmark this collaborative approach on Jigsaw level-4, which demands both spatial reasoning and rigorous verification, as shown in Figure 6(b). The results demonstrate that this partnership achieves superior accuracy, outperforming either model in isolation, and the performance further

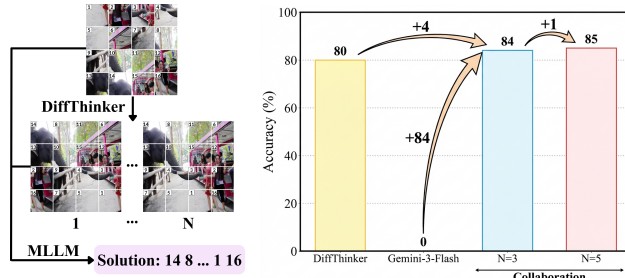

(a) Collaborative Pipeline  (b) Accuracy on Jigsaw level-4

*Figure 6.* **DiffThinker as a collaborative partner.** (a) The partnership framework where DiffThinker generates $N$ candidates for MLLM verification. (b) Performance on Jigsaw level-4, demonstrating that collaboration surpasses individual models and accuracy further scales with the number of candidates $N$.

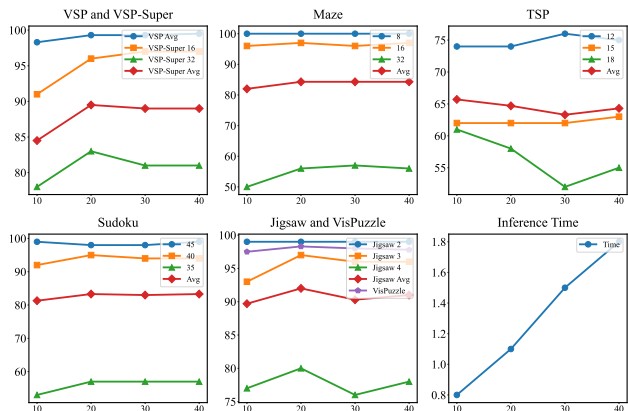

*Figure 7.* **Trade-off between accuracy and inference time across varying inference steps.** The horizontal axis denotes the number of inference steps, while the vertical axis denotes accuracy or inference time. An optimal balance between reasoning performance and computational cost is achieved at approximately 20 steps.

scales as the number of candidates increases. Specifically, DiffThinker compensates for the MLLM's limited visual imagination in spatial reasoning, while the MLLM leverages its reflective capabilities to filter potential errors in the generated candidates. This synergy reveals that DiffThinker can serve as a powerful visual reasoning backend to augment the cognitive breadth of MLLMs.

### 4.3. Ablation Studies

**Ablation on Inference Steps.** We first investigate the trade-off between accuracy and inference time, as illustrated in Figure 7. DiffThinker demonstrates remarkable robustness, maintaining high performance even with as few as 10 inference steps. Increasing the step count to 20 yields a noticeable performance boost, identifying an optimal balance between solution quality and computational efficiency. Beyond 20 steps, the accuracy plateaus with only marginal fluctuations, suggesting that the underlying reasoning mani-

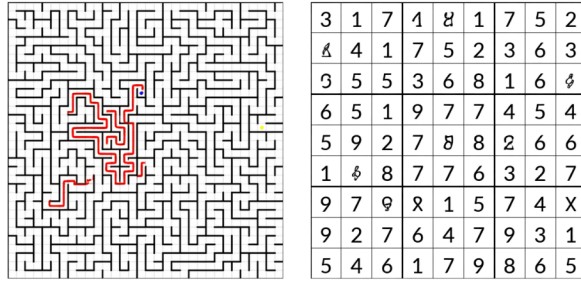

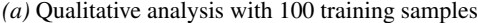

*(a)* Qualitative analysis with 100 training samples.

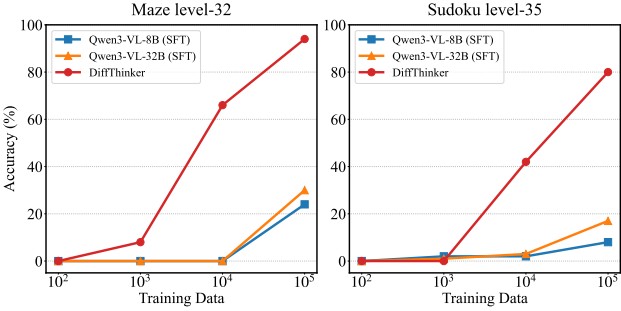

*(b)* Quantitative analysis with increasing training samples.

*Figure 8.* **Ablation on Training Data Scale.** (a) Qualitative analysis shows that with limited data, DiffThinker focuses on mastering rendering syntax. (b) Quantitative results on Maze level-32 and Sudoku level-35 demonstrate that DiffThinker scales consistently with data expansion.

fold is effectively captured early in the generative process. Based on these observations, we adopt 20 inference steps as our default configuration for evaluations to ensure superior performance with minimal inference overhead.

**Ablation on Training Data Scale.** We evaluate the influence of training data size on DiffThinker's performance using our most complex tasks, Maze level-32 and Sudoku level-35. We first qualitatively analyze the model's behavior under low-data regimes. As illustrated in Figure 8(a), due to the limited zero-shot reasoning capacity of the base model, DiffThinker initially focuses on mastering task-specific rendering syntax, such as grid alignment and trajectory continuity. As the training volume increases, DiffThinker transitions from superficial visual imitation to deep structural reasoning. Quantitative results in Figure 8(b) show that DiffThinker continues to benefit from data expansion, maintaining a consistent upward trajectory. With $10^5$ samples, the model effectively internalizes underlying causal structures, achieving over 90% accuracy on Maze level-32, while the performance of MLLMs remain significantly limited despite the increased data. Based on these observations, we utilize a total of 30,000 samples across all difficulty levels for each task in our main experiments to achieve an optimal balance between performance and efficiency.

**Ablation on Classifier-Free Guidance Scale.** We inves-

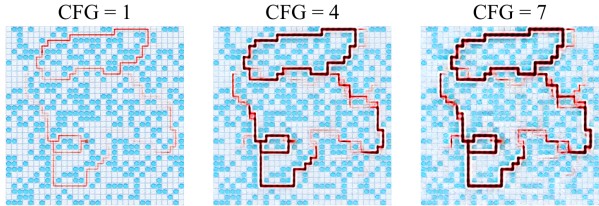

*(a)* Qualitative results of the predicted sample $\hat{x}_0$ at step 1.

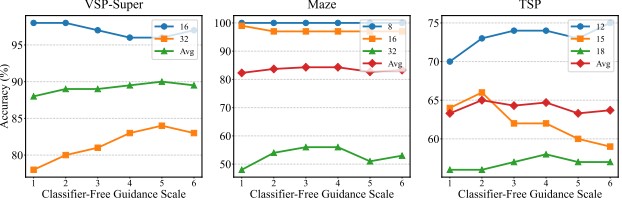

*(b)* Quantitative results of accuracy relative to CFG scales.

*Figure 9.* **Ablation on Classifier-Free Guidance scale.** (a) Impact of CFG scales on path clarity. (b) Accuracy trends across tasks confirming $w = 4$ as the peak performance point.

tigate the impact of Classifier-Free Guidance (CFG) (Ho & Salimans, 2022) on the reasoning capabilities of DiffThinker. As a core mechanism in diffusion models, CFG regulates the trade-off between conditional adherence and sample fidelity. The guided velocity field $\hat{v}_\theta$ is defined as:

$$\hat{v}_\theta(x_t, t, h) = v_\theta(x_t, t, \emptyset) + w(v_\theta(x_t, t, h) - v_\theta(x_t, t, \emptyset)) \quad (9)$$

where $v_\theta(x_t, t, h)$ and $v_\theta(x_t, t, \emptyset)$ represent the conditional and unconditional velocity predictions, respectively, and $w$ denotes the CFG scale. We begin with a qualitative assessment of different CFG scales. Figure 9(a) visualizes the predicted original sample $\hat{x}_0$ at the first step across varying scales. At $w = 1$, the insufficient conditioning produces faint and tentative trajectories, lacking the deterministic confidence for logical precision. Conversely, $w = 7$ triggers numerical over-saturation and visual artifacts, leading to distorted textures that degrade generative fidelity. Between these extremes, $w = 4$ acts as a logic amplifier, generating bold and precise paths that perfectly align with constraints.

Quantitatively, Figure 9(b) demonstrates that reasoning performance is robust across various guidance scales, with accuracy peaking at $w = 4$ across the majority of levels. Consequently, we adopt $w = 4$ as the default configuration for all experiments to ensure an optimal balance between logical precision and generative fidelity.

### 4.4. Image Generation vs. Video Generation.

Video generation offers unique advantages for multimodal reasoning by explicitly modeling temporal coherence and the continuous evolution of state transitions. Its capacity to represent reasoning trajectories as a fluid sequence could naturally facilitate the resolution of complex planning

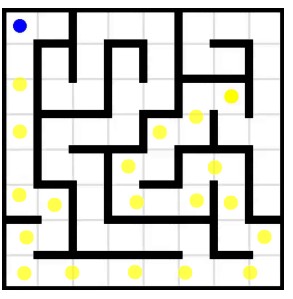

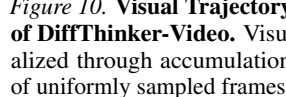

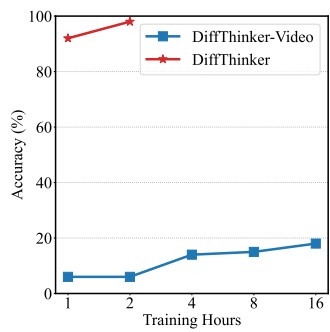

*Figure 10.* **Visual Trajectory of DiffThinker-Video.** Visualized through accumulation of uniformly sampled frames.

*Figure 11.* **Performance comparison between two paradigms.**

tasks. Therefore, we also explore the feasibility of video-based reasoning and conduct a direct comparison with our image-based approach. Our video-based baseline, denoted as DiffThinker-Video, is implemented upon Wan2.2-TI2V-5B (Wan et al., 2025), a leading open-source video foundation model. Due to the relatively weaker reasoning proficiency observed in current video generation models, we perform training and evaluation on Maze level-8, a relatively simple task that is also well-suited for video-based reasoning. To ensure a fair comparison, we train both models on identical datasets for varying numbers of epochs and report training duration and corresponding accuracy.

Qualitatively, Figure 10 demonstrates that DiffThinker-Video possesses inherent reasoning capabilities; it solves the maze problem by generating a video where a yellow ball progressively navigates the paths toward the target. Quantitatively, however, Figure 11 reveals that it yields lower accuracy with higher training overhead than DiffThinker. Furthermore, despite its smaller parameter count, DiffThinker-Video requires 2.0s per inference, nearly doubling the 1.1s latency of DiffThinker. These results highlight the prohibitive computational costs of video generation, underscoring the need for more powerful and efficient video models to advance generative multimodal reasoning.

## 5. Conclusion

In this paper, we introduce DiffThinker and establish Generative Multimodal Reasoning as a novel paradigm for complex vision-centric tasks. By leveraging diffusion models, we reformulate multimodal reasoning from a traditional text-centric symbolic mapping into a native generative image-to-image task, where the iterative denoising trajectory serves as a visual reasoning path, enabling the model to visualize and track the evolution of visual information throughout the reasoning process. Extensive experiments across four domains (sequential planning, combinatorial optimization, constraint satisfaction, and spatial configuration) demonstrate that DiffThinker significantly outperforms state-of-the-art MLLMs.

Our systematic analysis further reveals the intrinsic advantages of this paradigm, including its efficiency, controllability, and native parallelism, while showcasing its potential as a collaborative backend to augment the cognitive breadth of MLLMs. We hope DiffThinker will inspire further exploration into Generative Multimodal Reasoning to unlock the full potential of multimodal intelligent agents.

## Impact Statement

This work introduces a novel paradigm for multimodal reasoning. By reformulating reasoning as a generative process in visual space, our approach provides more stable solutions for applications such as visual planning and spatial configuration. While this framework offers significant benefits for decision making systems, we acknowledge that generative models can be misused to produce misleading visual information or misinformation. Furthermore, the computational resources required for training such models contribute to environmental considerations. We encourage researchers to deploy this technology responsibly and ensure that generative reasoning is used within ethical boundaries.

## Acknowledgement

We extend our gratitude to all the reviewers for their valuable feedback and suggestions, which greatly contributed to enhancing the quality of the paper. This work was supported by Shanghai Artificial Intelligence Laboratory.

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

# A. More Results and Analysis

## A.1. Analysis of Qwen3-VL

As illustrated in Table 4 and Figure 5, we observe that Qwen3-VL exhibits suboptimal zero-shot performance, whereas SFT yields consistent performance gains. Furthermore, by training the model to produce precise and structured outputs, SFT also effectively reduces inference latency. Notably, we find that GRPO demonstrates limited effectiveness on these complex long-horizon, vision-centric tasks. Due to the weak zero-shot capabilities of Qwen3-VL, directly employing a strict binary reward based on exact matching leads to unstable training trajectories and negligible improvements. Consequently, we incorporate partial reward functions in our experiments. While this facilitates the feasibility of GRPO training and results in certain performance enhancements, the inherent misalignment between the reward signals and the final evaluation objectives limits the potential for GRPO to further improve upon the SFT baseline.

## A.2. Comparison with "Thinking with Image" MLLMs

To further validate the effectiveness of the proposed Generative Multimodal Reasoning paradigm, we compare DiffThinker with several "Thinking with Image" MLLMs. Following the taxonomy in (Su et al., 2025b), our selection of baselines encompasses both tool-based approaches and those grounded in internal mental imagery. We select VSP and VisPuzzle as representative benchmarks due to their relative simplicity, while further incorporating VSP-Super to evaluate the scalability of models as the level of difficulty increases. Detailed results are summarized in Table 2.

On VSP and VSP-Super, "Thinking with Image" MLLMs are capable of solving low-difficulty scenarios; however, their performance deteriorates rapidly as the complexity increases. On VisPuzzle, DiffThinker outperforms ThinkMorph despite not having been specifically trained on corresponding training dataset. These results underscore the inherent advantages of DiffThinker over "Thinking with Image" MLLMs, particularly regarding its superior scalability to complex tasks.

*Table 2.* Comparison between DiffThinker and "Thinking with Image" MLLMs on VSP, VSP-Super, and VisPuzzle. Shaded cells indicate in-distribution evaluations, while * represents result evaluated by us.

| Model | VSP | | | | | | VSP-Super | | VisP. |
|---|---|---|---|---|---|---|---|---|---|
| | 3 | 4 | 5 | 6 | 7 | 8 | 16 | 32 | |
| DiffThinker | 99 | 100 | 98 | 99 | 100 | 100 | 96 | 83 | 98.3 |
| DeepEyes (Zheng et al., 2025) | 38* | 26* | 19* | 8* | 6* | 6* | 0* | 0* | 27.5* |
| ThinkMorph (Gu et al., 2025) | 90 | 84 | 83 | 75 | 66 | 57 | 0* | 0* | 79.0 |
| Anole (Chern et al., 2024) | 2 | 1 | 0 | 0 | - | - | - | - | - |
| MVoT (Li et al., 2025) | 21 | 11 | 8 | 3 | - | - | - | - | - |
| Mirage (Yang et al., 2025b) | 93 | 83 | 76 | 51 | - | - | - | - | - |

## A.3. Training Across Domains

Following previous studies (Xu et al., 2025; Li et al., 2025; Yang et al., 2025a), our primary experiments involve training independent models for VSP/VSP-Super, Maze, TSP, Sudoku, and Jigsaw, respectively. In this section, we also explore the feasibility of joint training by mixing datasets from different tasks, specifically encompassing VSP, VSP-Super, and Maze. The results comparing the independent models with this combined approach are presented in Table 3. Our results indicate that the joint training approach does not yield significant enhancements in performance; instead, we observe only marginal fluctuations in accuracy across the evaluated tasks. This outcome may be attributed to the inconsistent nature of the constraints that define each specific domain.

*Table 3.* Comparison of results between models trained on independent tasks and those trained across multiple domains simultaneously.

| Setting | VSP | | | | | | VSP-Super | | Maze | | |
|---|---|---|---|---|---|---|---|---|---|---|---|
| | 3 | 4 | 5 | 6 | 7 | 8 | 16 | 32 | 8 | 16 | 32 |
| Independent Training | 99 | 100 | 98 | 99 | 100 | 100 | 96 | 83 | 100 | 97 | 56 |
| Joint Training | 100 | 99 | 97 | 98 | 99 | 99 | 100 | 82 | 98 | 93 | 50 |

*Table 4.* **Detailed statistics for training and testing datasets across five task categories.**

| Task Category | Difficulty Levels | Training Samples | Test Samples |
|---|---|---|---|
| **VSP & VSP-Super** | 3, 4, 5, 6 (Grid size) | 500, 1,000, 2,500, 6,000 | 100 per level * |
| | 7, 8 (Grid size) | – | 100 per level [†] |
| | 16, 32 (Grid size) | 10,000, 10,000 | 100 per level |
| **Maze** | 8, 16, 32 (Grid size) | 10,000 | 100 per level |
| **TSP** | 12, 15 (City count) | 5000 | 100 per level |
| | 13, 14, 16, 17 (City count) | 5000 | – |
| | 18 (City count) | – | 100 per level[†] |
| **Sudoku** | 30 (Number of given clues) | 7,500 | – |
| | 35, 40, 45 (Number of given clues) | 7,500 | 100 per level |
| **Jigsaw & VisPuzzle** | $1\times2, 1\times3, 2\times1, 3\times1$ (Patch layout) | 4000 per level | – |
| | $2\times2, 3\times3, 4\times4$ (Patch layout) | 4,000, 5,000, 5,000 | 100 per level |
| | VisPuzzle | – | 400*[†] |

* denotes tasks utilizing official benchmarks from prior works.  [†] denotes out-of-distribution testing scenarios.

*Table 5.* **Hyperparameter settings for different training paradigms.**

| | Flow Matching | SFT | GRPO |
|---|---|---|---|
| Framework | DiffSynth-Studio (ModelScope, 2025) | SWIFT (Zhao et al., 2025) | verl (Sheng et al., 2024) |
| Epochs | 5 | 5 | 1 |
| Learning Rate | $1 \times 10^{-4}$ | $1 \times 10^{-4}$ | $1 \times 10^{-6}$ |
| LoRA Rank | 32 | 32 | – |
| Batch Size | 8 | 32 | 128 (8B) / 64 (32B) |
| Rollout Size ($n$) | – | – | 4 |
| KL Coefficient | – | – | $1 \times 10^{-2}$ |

# B. Implementation Details

## B.1. Training Details

### B.1.1. DATA PREPARATION.

The datasets utilized for training and evaluation are detailed in Table 4. Following previous research (Wang et al., 2025d; Wu et al., 2025c), we utilize the COCO (Lin et al., 2014) dataset to synthesize samples for both the training and testing of jigsaw puzzles. Specifically, we instantiate five independent models, each specialized for one of the five task categories, and subsequently evaluate them on their respective test benchmarks. All training datasets undergo thorough deduplication. Both DiffThinker and the baseline MLLMs are trained on identical data distributions to ensure an equitable comparison.

### B.1.2. HYPERPARAMETER CONFIGURATION.

We summarize the key training configurations and hyperparameters for Flow Matching, SFT, and GRPO in Table 5. In accordance with common practices, we employ Low-Rank Adaptation (LoRA) (Hu et al., 2022) for both the fine-tuning of Qwen-Image-Edit and the SFT of Qwen3-VL. For GRPO, considering the substantial computational overhead associated with reinforcement learning, we limit the training to a single epoch and utilize a reduced rollout number to maintain a manageable training budget while ensuring comparability across different experimental settings. For DiffThinker, we set the image resolution to $512 \times 512$ during both training and inference to ensure efficiency.

### B.1.3. REWARD FUNCTIONS FOR GRPO

Due to the limited zero-shot accuracy of the Qwen3-VL baselines on complex reasoning tasks, employing a strict binary reward based on exact matching results in extremely sparse signals, which significantly hinders the policy optimization process. Therefore, we design task-specific partial reward functions for each domain as follows:

**Sequential Planning (VSP, VSP-Super, and Maze).** For navigation tasks, we utilize a prefix matching reward. The reward

evaluates the longest continuous sequence of correct actions from the starting point. Given a predicted action sequence $P = (p_1, p_2, \ldots, p_m)$ and the ground truth $G = (g_1, g_2, \ldots, g_n)$, the reward is defined as:

$$R_{\text{plan}} = \frac{\max\{k \mid \forall i \leq k, p_i = g_i \text{ and } k \leq \min(m, n)\}}{n}. \tag{10}$$

**Combinatorial Optimization (TSP).** For the Traveling Salesperson Problem, the reward is designed to account for both coordinate set consistency and path length precision. Let $\mathcal{S}_p$ and $\mathcal{S}_g$ denote the sets of coordinates in the predicted and ground truth paths, and $L(\cdot)$ represent the total Euclidean distance of a trajectory. The reward is formulated as follows:

$$R_{\text{TSP}} = \begin{cases} 0.5\left(1 + \mathbb{I}(|L(P) - L(G)| < \epsilon)\right) & \text{if } \mathcal{S}_p = \mathcal{S}_g \\ 0 & \text{otherwise} \end{cases}, \tag{11}$$

where $\epsilon = 1 \times 10^{-4}$ serves as the tolerance for floating point comparisons.

**Constraint Satisfaction (Sudoku).** For Sudoku, if the length of the predicted sequence $|P|$ matches the standard 81 digits required for a $9 \times 9$ grid, the reward is calculated as the proportion of correctly filled cells. The reward function is defined as:

$$R_{\text{Sudoku}} = \begin{cases} \frac{1}{81} \sum_{i=1}^{81} \mathbb{I}(p_i = g_i) & \text{if } |P| = 81 \\ 0 & \text{otherwise} \end{cases}, \tag{12}$$

where $\mathbb{I}(\cdot)$ denotes the indicator function and $g_i$ represents the ground truth value for the $i$-th cell.

**Spatial Configuration (Jigsaw and VisPuzzle).** For Jigsaw tasks, if the length of the predicted sequence $|P|$ equals the total number of patches $n$, the reward is defined as the proportion of patches assigned to their correct absolute positions:

$$R_{\text{Jigsaw}} = \begin{cases} \frac{1}{n} \sum_{i=1}^{n} \mathbb{I}(p_i = g_i) & \text{if } |P| = n \\ 0 & \text{otherwise} \end{cases}. \tag{13}$$

## B.2. Prompt

Figures 12 through 17 provide a comprehensive overview of the prompt templates utilized in our study. For VSP and VSP-Super, we adopt the original prompt (Wu et al., 2024) designs as specified in the primary literature for the evaluation of MLLMs. However, for SFT, the prompt structures are specifically adapted as illustrated in Figure 13. This modification is necessitated by the fact that our SFT paradigm does not employ Chain-of-Thought (CoT), requiring a more direct and concise instructional format to ensure consistency with the supervised training objectives.

---

*VSP and VSP-Super:*
Draw a continuous red line connecting the Start point to the Goal point, avoiding all holes.
*Maze:*
Draw a continuous red line connecting the yellow dot to the blue dot, avoiding all walls.
*TSP:*
Connect all points with red line segments to form the shortest closed loop.
*Sudoku:*
Solve this Sudoku puzzle.
*Jigsaw and VisPuzzle:*
Solve this Jigsaw puzzle.

---

*Figure 12.* **Prompt templates for DiffThinker.**

## C. Limitations and Future Work

DiffThinker demonstrates state-of-the-art performance in vision-centric reasoning within targeted domains. However, its out-of-distribution generalization remains constrained by the limited zero-shot reasoning proficiency of current generative foundation models. Future research should prioritize the development of more robust multimodal generative foundation models specifically optimized for reasoning. Building upon such foundations, we aim to further explore the boundaries of generative multimodal reasoning and enhance its capability to generalize across broader, out-of-distribution tasks.

Furthermore, this work primarily focuses on vision-centric tasks, where DiffThinker significantly surpasses MLLMs. However, it is important to acknowledge that MLLMs maintain a clear advantage in text-centric domains, such as complex mathematical problems. A promising future direction lies in investigating deeper collaboration and synergy between generative multimodal reasoners and MLLMs. By integrating the superior visual imagination capability of DiffThinker with the advanced linguistic and symbolic capabilities of MLLMs, we can extend the scope of multimodal reasoning to a wider spectrum of diverse and demanding tasks.

## D. Qualitative Analysis

To facilitate a better understanding of the performance disparities between DiffThinker and MLLMs, we provide success and failure cases of DiffThinker for each task, along with the thinking processes of Gemini-3-Pro (Google, 2025a), as shown in Figures 18 through 35. We utilize Google AI Studio to evaluate Gemini-3-Pro and obtain its reasoning duration. For each task, we evaluate Gemini-3-Pro on the same problem instances where DiffThinker achieved successful solutions.

As a professional maze solver, your task is to analyze a grid-based map and devise an action plan that enables a player to reach the goal from the starting point without falling into any holes, using the fewest possible moves. Since coding is not within your skill set, your approach relies on logical reasoning of the map.

## Game Setup
- The game presents a fully observable grid-based map.
- The player starts at a specified grid square, with the goal located elsewhere on the map.
- Each grid square is either safe or contains a hole.
- Your goal is to guide the player to the goal while avoiding holes.
The following figure shows how the player, the holes (non-safe grid), the lands (safe grids), and the goals look like.

<Image>

## Moving Rules
- The action plan involves a series of moves: 'L' (left), 'R' (right), 'U' (up), or 'D' (down).
- Each move transfers the player to the adjacent square in that direction, provided it is a safe square. The player cannot move more than one square at a time.
- Moving off the edge of the map has no effect. The player will remain at the same square.
- DO NOT MOVE INTO A HOLE! Falling into a hole results in defeat.
- Locating at the grid containing the goal results in victory.
We provide an example to further illustrate the rules.

<Image>

In this provided example:
- The player is at Row 1, Column 1;
- The goal is at Row 4, Column 4;
- There are two holes: one at Row 1, Column 2, and another at Row 4, Column 1.
- The player can move DOWN. This is because moving down brings them to Row 2, Column 1, and this cell is safe (without holes).
- Moving UP has no effects. This is because the player is already in the topmost row.
- Similarly, moving LEFT has no effects because the player is already in the left-most column.
- Moving RIGHT places the player at Row 1, Column 2. Since there is a hole at this grid, this move results in a loss.

(For SFT)
## Procedure and Output
Now you will solve the given maze. To solve it, please output an aggregated plan using "Action plan: <PLAN>", where <PLAN> is a string concatenated action in each step. For example, "Action plan: L,L,R,U,D" meaning an action plan of left, left, right, up, and down.
Do not output any extra content after the above aggregated output.

Please generate action plan for the following maze:

(For GRPO)
## Procedure and Output
Now you will solve the given maze. To solve it, please output an aggregated plan using "Action plan: <PLAN>", where <PLAN> is a string concatenated action in each step. For example, "Action plan: L,L,R,U,D" meaning an action plan of left, left, right, up, and down.
Do not output any extra content after the above aggregated output.

Please generate action plan for the following maze:
To solve it, please generate text EXACTLY FOLLOW THE FOLLOWING STEPS:
1. First, interpret map. List where the player is at now, where is the goal, and where are the holes.
2. Then, generate an action plan to navigate to the goal step by step. At each step, you should check:
 (a) Where the current move leads the player to (the row and column);
 (b) What is in that grid. Is it a hole? Is it the goal? Is it an empty space?
 (c) Determine if that is a safe action. If not, correct it and re-generate the action plan.
3. Next, verify if the steps successfully navigate the player to the goal without falling into the hole. If not, restart from step 2 and re-generate this step.
4. If succeed, output an aggregated plan using "Action plan: <PLAN>", where <PLAN> is a string concatenated action in each step. For example, "Action plan: L,L,R,U,D" meaning an action plan of left, left, right, up, and down. Double check the final action plan is consistent with the previous analysis.
Do not output any extra content after the above aggregated output.

Please generate action plan for the following maze:

Image1 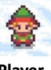 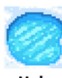 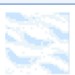 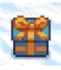
Player     Hole (Not safe)    Land (Safe)    Goal

Image2 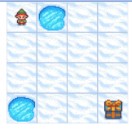

*Figure 13.* **Prompt templates of VSP and VSP-Super for MLLMs.**

You are required to solve this maze problem. The yellow dot is the starting point, and the blue dot is the ending point. You may (or may not) think before answering. Your response must end with "Action plan: <PLAN>", where <PLAN> is a string of concatenated actions for each step. For example, "Action plan: L,L,R,U,D" means an action plan of left, left, right, up, and down. Do not output any extra content after the above aggregated output.

*Figure 14.* **Prompt templates of Maze for MLLMs.**

You are required to solve this Traveling Salesperson Problem (TSP). The yellow dot is the starting point, and the blue dots are the cities that must be visited exactly once.

The image corresponds to a 0-indexed grid coordinate system:
1. The top-left corner is (0,0).
2. Coordinates are formatted as (x,y), where x is the horizontal index (column) and y is the vertical index (row).

You need to find the shortest closed loop. You may (or may not) think before answering. Your response must end with "Path: <PATH>", where <PATH> is a string of coordinate tuples representing the visiting order (e.g., "(0,0),(2,3),(5,5),(0,0)"). Do not output any extra content after the above aggregated output.

*Figure 15.* **Prompt templates of TSP for MLLMs.**

You are required to solve this 9x9 Sudoku puzzle. You may (or may not) think before answering. Your response must end with "Solution: <SOLUTION_STRING>", where <SOLUTION_STRING> is a single string of 81 digits representing the entire solved grid, read from left to right, top to bottom. Do not output any other text after this.

*Figure 16.* **Prompt templates of Sudoku for MLLMs.**

You are required to solve this jigsaw puzzle. The image consists of shuffled blocks, each labeled with a number in the top-left corner. You need to determine the correct order to restore the original image. You may (or may not) think before answering. Your response must end with "Solution: <SOLUTION_STRING>", where <SOLUTION_STRING> is a space separated sequence of the block numbers corresponding to the restored image, read from left to right, top to bottom. Do not output any other text after this.

*Figure 17.* **Prompt templates of Jigsaw and VisPuzzle for MLLMs.**

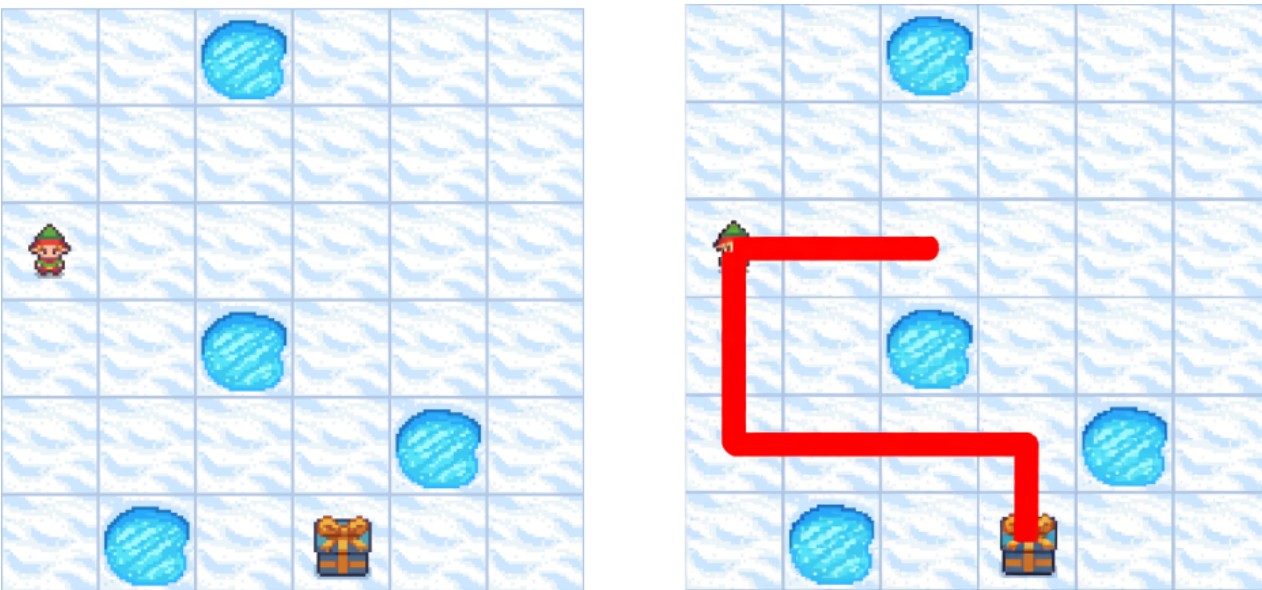

*Figure 18.* **Failure case of DiffThinker on VSP.** In a simple task, DiffThinker performs excessive parallel reasoning but fails to preserve a unique trajectory, ultimately leading to a failure.

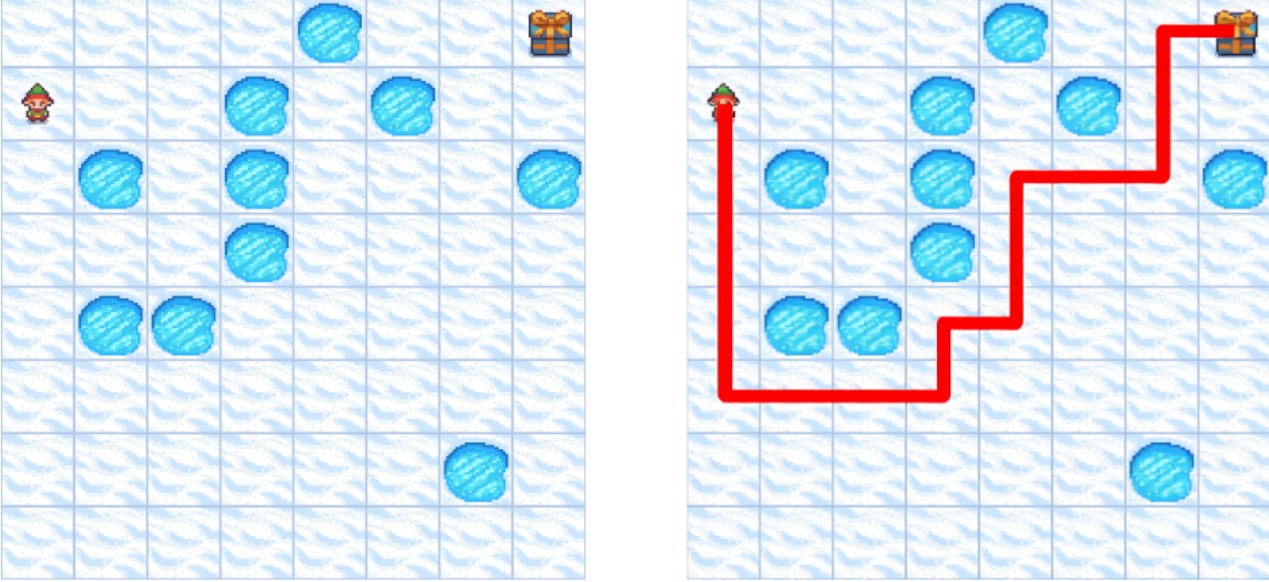

*Figure 19.* **Success case of DiffThinker on VSP.**

**(After thinking for 73.1s)**

Based on the grid analysis:

1.  **Starting Position:** The elf is at **Row 1, Column 0** (0-indexed from top-left).
2.  **Goal Position:** The gift is at **Row 0, Column 7**.
3.  **Obstacles (Holes):**
    *   Row 0: (0,4)
    *   Row 1: (1,3), (1,5)
    *   Row 2: (2,1), (2,3), (2,7)
    *   Row 3: (3,3)
    *   Row 4: (4,1), (4,2)
    *   Row 6: (6,6)
4.  **Pathfinding:**
    *   The **Upper Path** is blocked: Moving Right from start leads to holes at (1,3). maneuvering to Row 0 is blocked by the hole at (0,4).
    *   The **Middle Path** is blocked by a cluster of holes in Rows 2, 3, and 4.
    *   The **Lower Path** is the only viable option. We must travel down to **Row 5**, which is the first completely clear row spanning the width needed to bypass the left-side obstacles.
    *   Once on Row 5, we travel Right to **Column 6**. We cannot use Column 7 to ascend because of the hole at (2,7). Column 6 is clear from Row 5 up to Row 0 (the hole at (6,6) is below our path).
    *   From (0,6), we move Right to the goal at (0,7).

**Step-by-Step Execution:**
1.  **Move Down 4 times:** (1,0) $\to$ (2,0) $\to$ (3,0) $\to$ (4,0) $\to$ (5,0).
2.  **Move Right 6 times:** (5,0) $\to$ (5,1) $\to$ (5,2) $\to$ (5,3) $\to$ (5,4) $\to$ (5,5) $\to$ (5,6).
3.  **Move Up 5 times:** (5,6) $\to$ (4,6) $\to$ (3,6) $\to$ (2,6) $\to$ (1,6) $\to$ (0,6).
4.  **Move Right 1 time:** (0,6) $\to$ (0,7) [Goal].

Action plan: D,D,D,D,R,R,R,R,R,R,U,U,U,U,U,R

*Figure 20.* **Thinking process of Gemini-3-Pro on VSP.** Gemini-3-Pro successfully provides the correct solution.

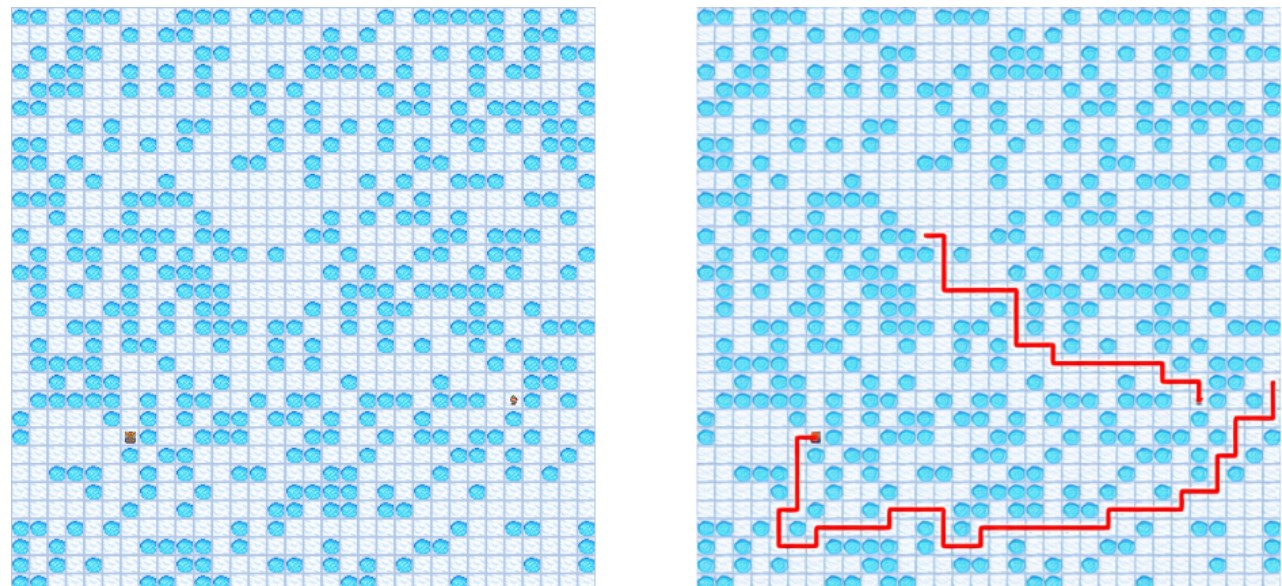

*Figure 21.* **Failure case of DiffThinker on VSP-Super.** In a complex task, DiffThinker identifies a nearly correct trajectory; however, the path is obstructed by a hole, preventing further progress and leading to an ultimate failure.

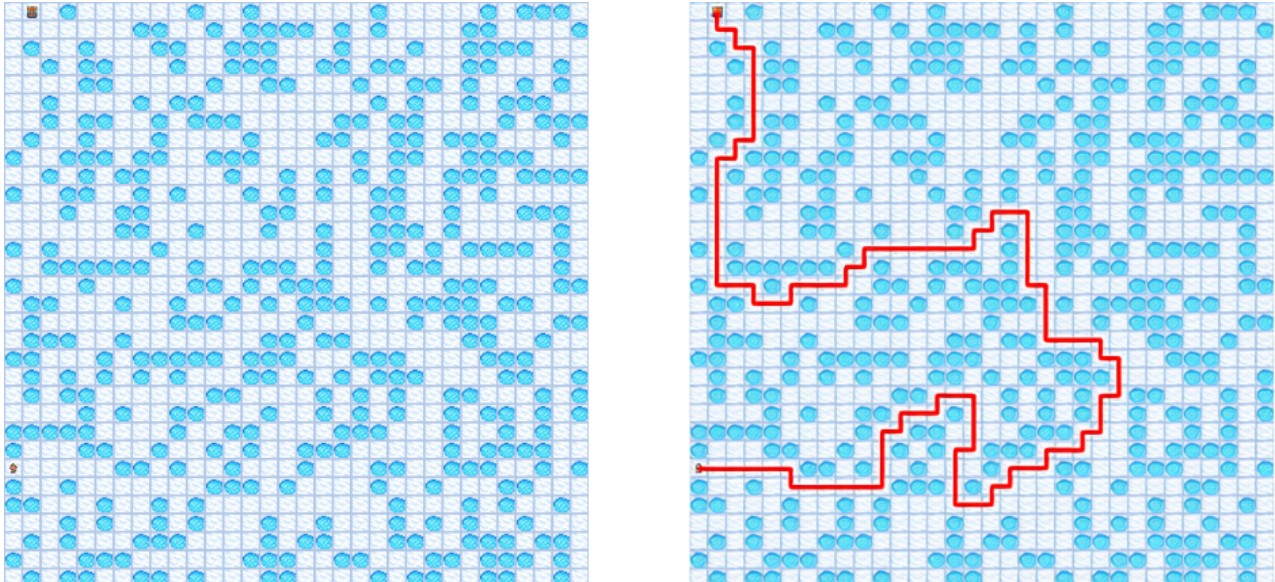

*Figure 22.* **Success case of DiffThinker on VSP-Super.**

**(After thinking for 164.4s)**

Based on a careful analysis of the grid, the starting position is at **(0, 25)** (column 0, row 25, counting from top-left (0,0)), and the goal is at **(0, 0)**. The path involves navigating the leftmost columns (0, 1, and 2) to avoid the specific placement of holes.

Here is the step-by-step logic:
1. **Start (0, 25):** The tile immediately above at (0, 24) is a hole. Move **Right** to (1, 25).
2. **Move Up:** Move **Up, Up** to (1, 23). The path ahead at (1, 22) is blocked by a hole.
3. **Bypass Hole (Row 22):** Move **Right** to (2, 23), then **Up, Up** to (2, 21).
4. **Return to Lane 1:** The path at (2, 19) is blocked by a hole, but (1, 19) is clear. Move **Left** to (1, 21).
5. **Move Up:** Move **Up 5 times** to reach (1, 16). The path ahead at (1, 15) is blocked by a hole.
6. **Bypass Hole (Row 15):** Move **Right** to (2, 16).
7. **Long Upward Stretch:** Stay in column 2 to avoid a hole at (1, 8). Move **Up 9 times** to reach (2, 7). The path ahead at (2, 6) is blocked by a hole.
8. **Return to Lane 1:** Move **Left** to (1, 7).
9. **Final Stretch:** Column 1 is clear to the top. Move **Up 7 times** to reach (1, 0).
10. **Goal:** Move **Left** to reach the goal at (0, 0).

Action plan: R,U,U,R,U,U,L,U,U,U,U,U,R,U,U,U,U,U,U,U,U,U,L,U,U,U,U,U,U,U,L

*Figure 23.* **Thinking process of Gemini-3-Pro on VSP-Super.** Gemini-3-Pro fails to provide the correct solution.

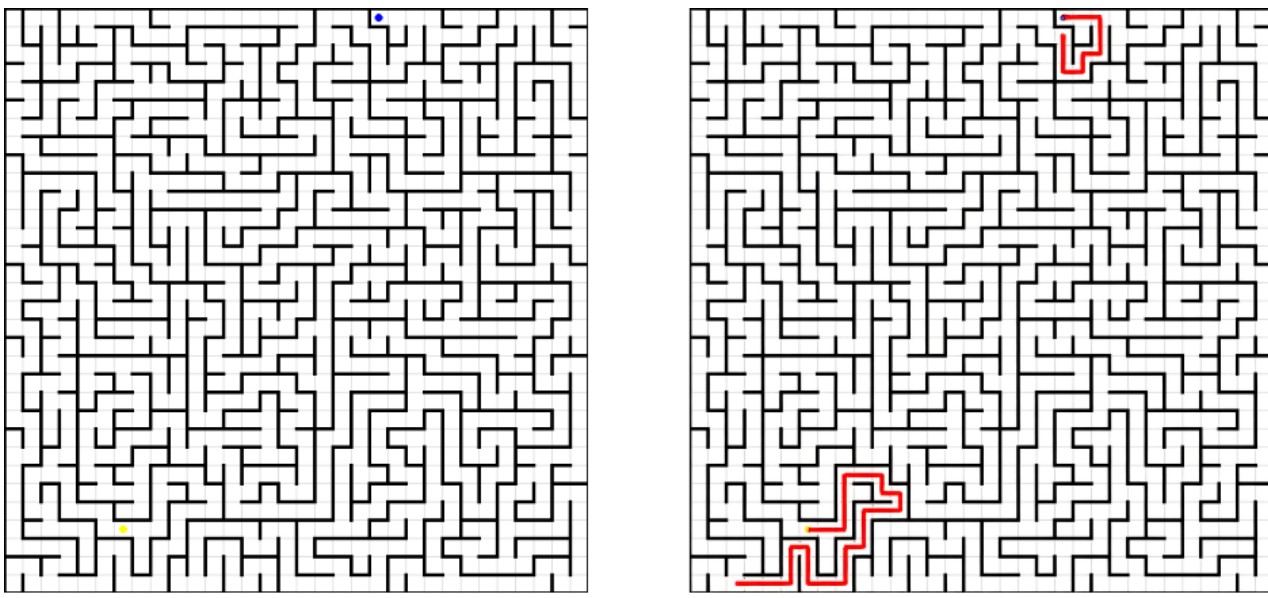

*Figure 24.* **Failure case of DiffThinker on Maze.** In an instance characterized by a significant distance between the starting point and the goal, DiffThinker fails to sustain deep reasoning and provides only a preliminary trajectory.

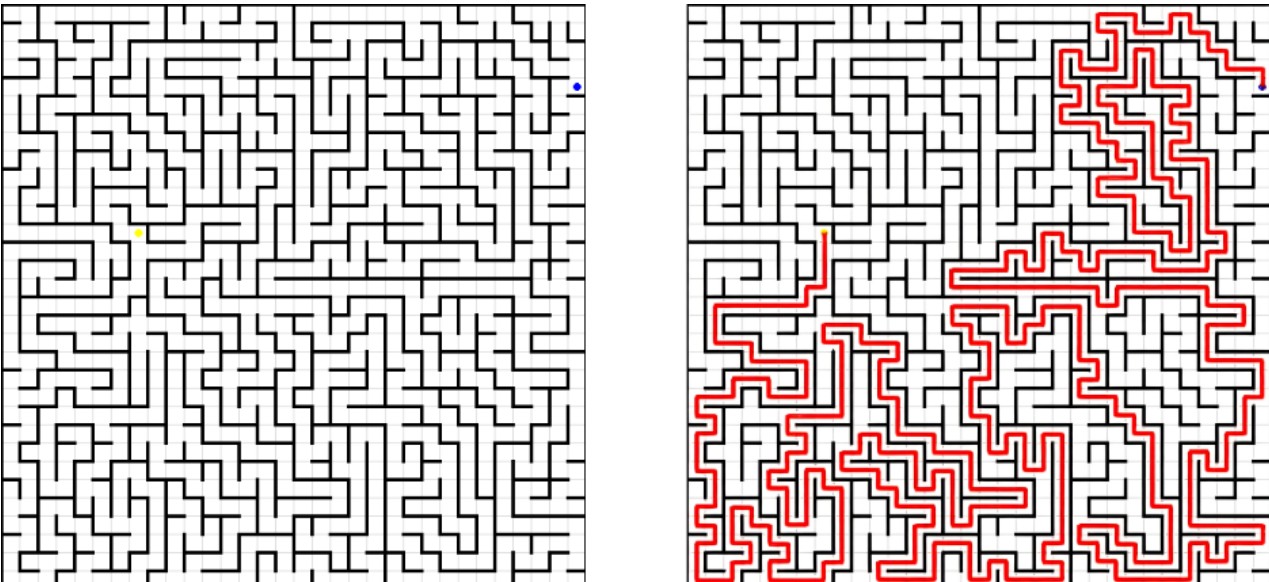

*Figure 25.* **Success case of DiffThinker on Maze.**

**(After thinking for 135.3s)**

To solve the maze, we first identify the starting point (yellow dot) and the ending point (blue dot).

**1. Coordinates:**
* **Start (Yellow Dot):** Located at approximately Row 12, Column 7 (0-indexed from top-left).
* **End (Blue Dot):** Located at approximately Row 4, Column 31 (top-right corner).

**2. Pathfinding Strategy:**
We will trace the path segment by segment.

* **Segment 1: Start to Upper Left Channel**
  * From the start (12, 7), the immediate path up and left is blocked. The path down leads to a dead end. We must go **Right**.
  * Move **Right 4 steps** to (12, 11).
  * Hit a wall, move **Up 1 step** to (11, 11).
  * Right is blocked, so move **Left 4 steps** to (11, 7).
  * Now we are in a long vertical channel. Move **Up 7 steps** to (4, 7).

**(Intermediate portions of the content are omitted due to excessive length.)**

* **Segment 5: The Deep Detour**
  * At (4, 21), the path right is blocked. We must go around the obstacle below.
  * Move **Down 3 steps** to (7, 21).
  * Move **Right 1 step** to (7, 22).
  * Move **Up 2 steps** to (5, 22).
  * Move **Right 1 step** to (5, 23).
  * Move **Up 1 step** to (4, 23).

* **Segment 6: Final Stretch**
  * We are now back on Row 4 with a clear path to the blue dot.
  * Move **Right 8 steps** from (4, 23) to the goal at (4, 31).

**Aggregated Moves:**
R,R,R,R, U, L,L,L,L, U,U,U,U,U,U,U, R,R, D, R,R, U,U,U,U,U, R,R,R, D,D,D, R,R,R, U,U,U, R,R,R, D,D, L, D,D, R,R, D,D,D, R, U,U, R, U, R,R,R,R,R,R,R,R

Action plan:
R,R,R,R,U,L,L,L,L,U,U,U,U,U,U,U,R,R,D,R,R,U,U,U,U,U,R,R,R,D,D,D,R,R,R,U,U,U,R,R,R,D,D,L,D,D,R,
R,D,D,D,R,U,U,R,U,R,R,R,R,R,R,R,R

*Figure 26.* **Thinking process of Gemini-3-Pro on Maze.** Gemini-3-Pro fails to provide the correct solution.

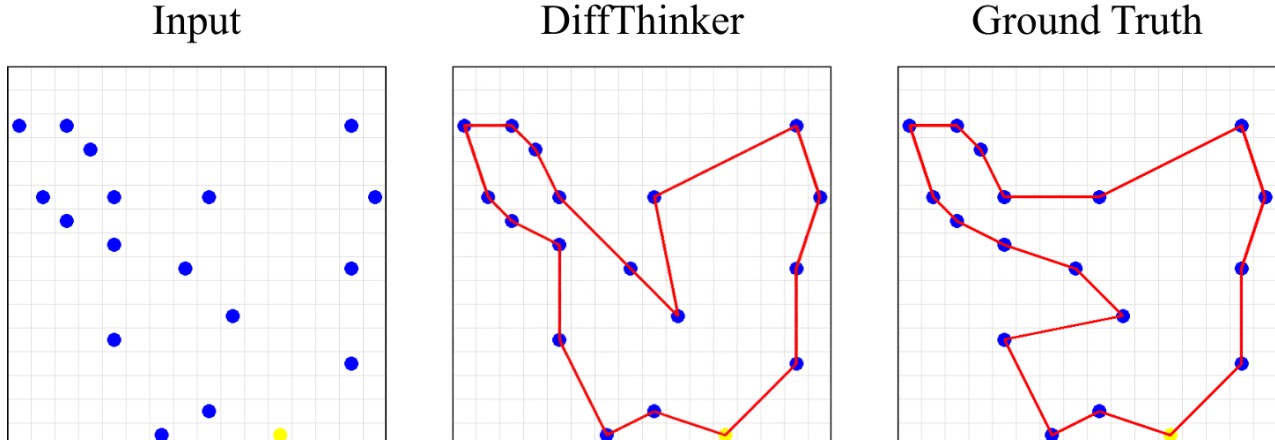

*Figure 27.* **Failure case of DiffThinker on TSP.** DiffThinker successfully identifies a feasible closed loop, yet it is not the shortest path.

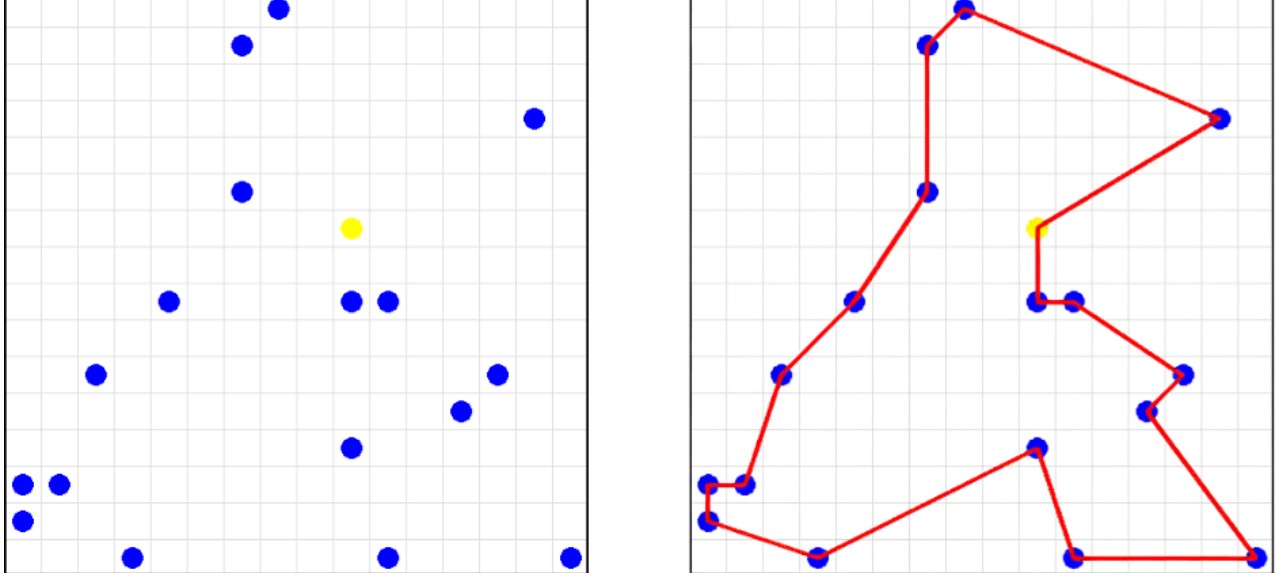

*Figure 28.* **Success case of DiffThinker on TSP.**

**(After thinking for 200.4s)**

To solve the Traveling Salesperson Problem for the given grid, we first identify the coordinates of all the points.

1. **Grid Setup:** 16x16 grid, 0-indexed.
2. **Points Identification:**
   * **Start (Yellow):** (9,6)
   * **Cities (Blue):**
      * Top: (7,0), (6,1), (14,3)
      * Upper Middle: (6,5), (4,8), (9,8), (10,8)
      * Right: (13,10), (12,11), (15,15)
      * Bottom: (10,15), (9,12), (3,15)
      * Bottom Left Cluster: (2,10), (1,13), (0,13), (0,14)

**Strategy:**
We will construct a path using a nearest-neighbor approach combined with visual optimization to minimize long jumps, effectively creating a loop that follows the perimeter and weaves through the central clusters.

1. **Start at (9,6).**
2. Go to the isolated top-right point **(14,3)**. (Distance: ~5.8)
3. Move to the top cluster: **(7,0)** then **(6,1)** then **(6,5)**.
4. Travel down the left side: **(4,8)** then **(2,10)**.
5. Navigate the bottom-left corner: **(1,13)** -> **(0,13)** -> **(0,14)** -> **(3,15)**.
6. Jump to the bottom-center/right: **(9,12)**.
7. Complete the bottom-right loop: **(10,15)** -> **(15,15)** -> **(12,11)** -> **(13,10)**.
8. Return to the center cluster: **(10,8)** -> **(9,8)**.
9. Return to Start: **(9,6)**.

**Path Construction:**
(9,6) -> (14,3) -> (7,0) -> (6,1) -> (6,5) -> (4,8) -> (2,10) -> (1,13) -> (0,13) -> (0,14) -> (3,15) -> (9,12) -> (10,15) -> (15,15) -> (12,11) -> (13,10) -> (10,8) -> (9,8) -> (9,6)

Path: (9,6),(14,3),(7,0),(6,1),(6,5),(4,8),(2,10),(1,13),(0,13),(0,14),(3,15),(9,12),(10,15),(15,15),(12,11), (13,10),(10,8),(9,8),(9,6)

*Figure 29.* **Thinking process of Gemini-3-Pro on TSP.** Gemini-3-Pro successfully provides the correct solution.

### Input

| | | | | | 5 | | | 2 |
|---|---|---|---|---|---|---|---|---|
| 2 | | 8 | | 7 | | | | 6 |
| | 7 | | | 6 | 4 | | 5 | 9 |
| | 9 | 7 | 6 | 1 | | 5 | | 8 |
| 3 | 6 | | 4 | 8 | 2 | | | |
| | 1 | | | | 7 | | 3 | |
| | | 1 | 5 | | 9 | | | |
| | 3 | 9 | | 4 | | | 8 | |
| | | 3 | 2 | | | 6 | | |

### DiffThinker

| 9 | 4 | 1 | 8 | 3 | 5 | 6 | 7 | 2 |
|---|---|---|---|---|---|---|---|---|
| 2 | 5 | 8 | 9 | 7 | 1 | 3 | 4 | 6 |
| 1 | 7 | 3 | 2 | 6 | 4 | 8 | 5 | 9 |
| 4 | 9 | 7 | 6 | 1 | 3 | 5 | 2 | 8 |
| 3 | 6 | 5 | 4 | 8 | 2 | 7 | 9 | 1 |
| 8 | 1 | 2 | 5 | 9 | 7 | 6 | 3 | 4 |
| 7 | 2 | 6 | 1 | 5 | 8 | 9 | 4 | 3 |
| 6 | 3 | 9 | 7 | 4 | 6 | 2 | 8 | 5 |
| 5 | 8 | 4 | 3 | 2 | 9 | 1 | 6 | 7 |

### Ground Truth

| 9 | 4 | 6 | 8 | 3 | 5 | 7 | 1 | 2 |
|---|---|---|---|---|---|---|---|---|
| 2 | 5 | 8 | 9 | 7 | 1 | 3 | 4 | 6 |
| 1 | 7 | 3 | 2 | 6 | 4 | 8 | 5 | 9 |
| 4 | 9 | 7 | 6 | 1 | 3 | 5 | 2 | 8 |
| 3 | 6 | 5 | 4 | 8 | 2 | 1 | 9 | 7 |
| 8 | 1 | 2 | 5 | 9 | 7 | 6 | 3 | 4 |
| 6 | 2 | 4 | 1 | 5 | 8 | 9 | 7 | 3 |
| 5 | 3 | 9 | 7 | 4 | 6 | 2 | 8 | 1 |
| 7 | 8 | 1 | 3 | 2 | 9 | 4 | 6 | 5 |

*Figure 30.* **Failure case of DiffThinker on Sudoku.** DiffThinker successfully populates most of entries, yet commits several errors.

| | 1 | 8 | 5 | | | 4 | 2 | 6 |
|---|---|---|---|---|---|---|---|---|
| 6 | 5 | 3 | | 8 | | 1 | 7 | |
| | 7 | | 9 | 1 | | 5 | 3 | |
| 2 | 3 | | | | | | | 5 |
| | 6 | 4 | 3 | 2 | | | 1 | |
| | 9 | | | | | | | |
| | | | | | | 6 | 8 | |
| | | 6 | | 7 | 2 | 3 | | |
| | | 1 | | 4 | | 7 | | |

| 9 | 1 | 8 | 5 | 3 | 7 | 4 | 2 | 6 |
|---|---|---|---|---|---|---|---|---|
| 6 | 5 | 3 | 2 | 8 | 4 | 1 | 7 | 9 |
| 4 | 7 | 2 | 9 | 1 | 6 | 5 | 3 | 8 |
| 2 | 3 | 7 | 4 | 9 | 1 | 8 | 6 | 5 |
| 8 | 6 | 4 | 3 | 2 | 5 | 9 | 1 | 7 |
| 1 | 9 | 5 | 7 | 6 | 8 | 2 | 4 | 3 |
| 7 | 2 | 9 | 1 | 5 | 3 | 6 | 8 | 4 |
| 5 | 4 | 6 | 8 | 7 | 2 | 3 | 9 | 1 |
| 3 | 8 | 1 | 6 | 4 | 9 | 7 | 5 | 2 |

*Figure 31.* **Success case of DiffThinker on Sudoku.**

**(After thinking for 158.9s)**

To solve the Sudoku puzzle, we analyze the grid and apply standard Sudoku deduction techniques (scanning rows, columns, and 3x3 boxes for missing numbers).

**Step-by-step resolution:**

1.  **Row 1:** Missing {3, 7, 9}. By cross-referencing columns and boxes, we determine R1C1=9, R1C5=3, R1C6=7.
2.  **Box 1 (Top-Left):** R1C1=9 leaves {2, 4} for R3C1 and R3C3. R3C3 must be 2 (due to 4 in column 3), so R3C1=4.
3.  **Box 2 (Top-Middle):** Remaining cells are filled. R2C4=2, R2C6=4, R3C6=6, R3C9=8.
4.  **Box 3 (Top-Right):** R2C9=9 becomes clear.
5.  **Middle Band (Rows 4-6):**
    *   Row 4: Missing {1, 4, 6, 7, 8, 9}. Logic on columns places R4C3=7, R4C4=4, R4C5=9, R4C6=1, R4C7=8, R4C8=6.
    *   Row 5: Missing {5, 7, 8, 9}. Constraints force R5C1=8, R5C6=5, R5C7=9, R5C9=7.
    *   Row 6: Missing {1, 2, 3, 4, 5, 6, 7, 8}. Constraints force R6C1=1, R6C3=5, R6C4=7, R6C5=6, R6C6=8, R6C7=2, R6C8=4, R6C9=3.
6.  **Bottom Band (Rows 7-9):**
    *   Columns are filled vertically.
    *   Col 1: R7C1=7, R8C1=5, R9C1=3.
    *   Col 2: R7C2=2, R8C2=4, R9C2=8.
    *   Col 3: R7C3=9.
    *   Col 4: R7C4=1, R8C4=8, R9C4=6.
    *   Col 5: R7C5=5.
    *   Col 6: R7C6=3, R9C6=9.
    *   Box 9 fills out with R7C9=4, R8C8=9, R8C9=1, R9C8=5, R9C9=2.

**Final Grid:**
```
9 1 8 | 5 3 7 | 4 2 6
```
**(Intermediate portions of the content are omitted due to excessive length.)**
```
3 8 1 | 6 4 9 | 7 5 2
```

Solution:
918537426653284179472916538237491865864325917195768243729153684546872391381649752

*Figure 32.* **Thinking process of Gemini-3-Pro on Sudoku.** Gemini-3-Pro successfully provides the correct solution.

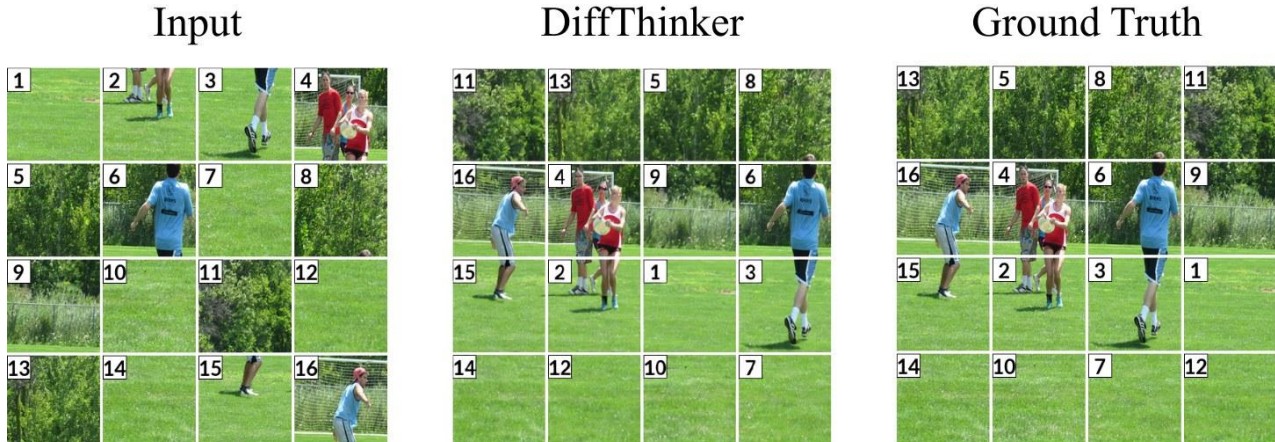

*Figure 33.* **Failure case of DiffThinker on Jigsaw.** Due to the fact that our test set is generated at random, certain instances contain regions that are extremely difficult to distinguish. DiffThinker produces a globally reasonable image, yet the fine details remain incorrect.

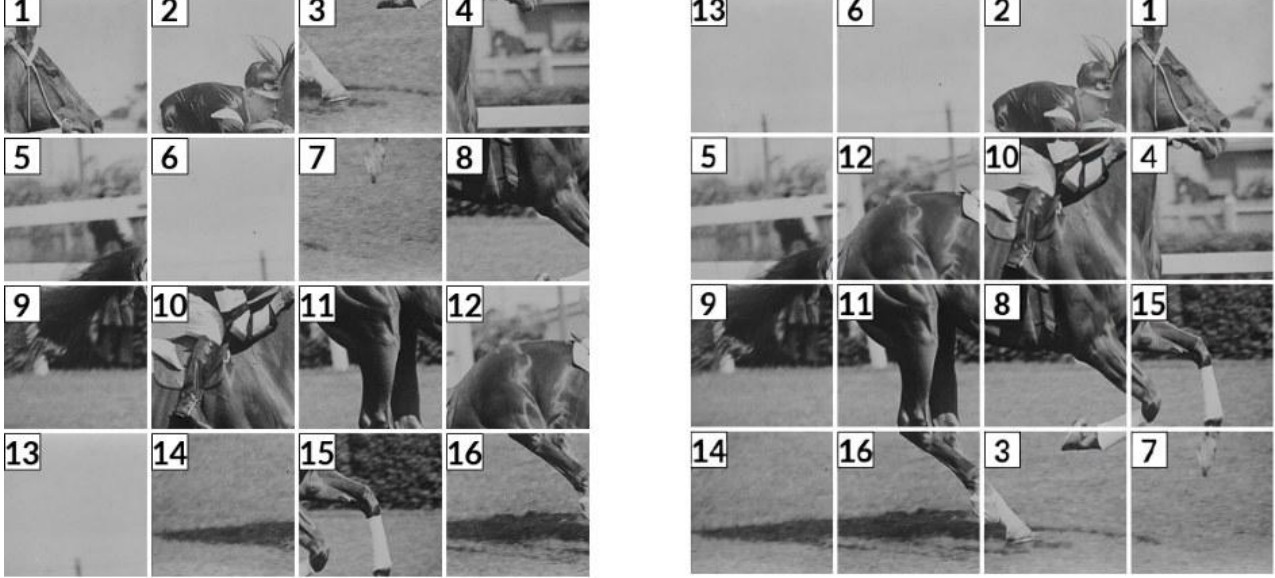

*Figure 34.* **Success case of DiffThinker on Jigsaw.**

**(After thinking for 131.7s)**

Based on the visual analysis of the image, the blocks are already arranged in the correct order to form a coherent picture of a racehorse and jockey in motion.

Here is the breakdown of the connections:
*   **Row 1 (Sky and Heads):** Block 13 (sky) and Block 6 (sky) form the background. Block 2 shows the jockey's head and back, which connects perfectly to Block 1 showing the horse's head and neck.
*   **Row 2 (Body and Fence):** Block 5 shows the background fence. Block 12 shows the horse's rump, connecting to Block 10 (saddle and jockey's leg), which connects to Block 4 (horse's chest and martingale). The lines of the horse's back and the jockey's position are continuous.
*   **Row 3 (Legs and Tail):** Block 9 shows the tail, connecting to the rear thigh in Block 11. Block 8 shows the belly, connecting to the front leg/chest area in Block 15. The musculature matches perfectly.
*   **Row 4 (Ground and Hooves):** Block 14 shows the shadow of the tail, connecting to the shadow/leg in Block 16. Block 3 shows the planted leg, and Block 7 shows the leading front hoof in the air, matching the leg positions from the row above.

The numbers on the blocks, read row by row, from left to right, are:

Solution: 13 6 2 1 5 12 10 4 9 11 8 15 14 16 3 7

*Figure 35.* **Thinking process of Gemini-3-Pro on Jigsaw.** Gemini-3-Pro successfully provides the correct solution.

