# OpenReview forum: "DiffThinker: Towards Generative Multimodal Reasoning with Diffusion Models"
_ICML.cc/2026/Conference — ICML 2026 regular_

### Official Review · Reviewer_Cpja · 2026-03-08

**Soundness:** 3
**Presentation:** 3
**Significance:** 3
**Originality:** 3
**Overall Recommendation:** 4
**Confidence:** 4

**Summary:**

This paper proposed DiffThinker, which reformulates some multimodal reasoning tasks as a generative text-to-image / image-to-image task by observing the denoising trajectory as the reasoning path. The continually trained model outperformed existing models and achieved highly accurate results in solving five vision-centric reasoning tasks.

**Compliance With Llm Reviewing Policy:**

Affirmed.

**Final Justification:**

The rebuttal has addressed my main concerns. I think the authors' response helps consolidate the emergent reasoning behavior of diffusion models.

**Key Questions For Authors:**

1. Please also see the weakness section. It seems that the flow matching training of the model is pretty standard and the inference is the standard first-order Euler solver. I'm curious about its connection with the models' reasoning capability. More importantly, I’m wondering if the authors could explain with more details why the claim that “the denoising trajectory $\{x_t\}$ naturally serves as a visual reasoning path, where each $\{x_t\}$ captures an intermediate state of the solution image” (line 195) can be guaranteed. I think more evidence is needed to support such a main claim in the paper, beyond a single qualitative example from a single task in Fig.4. I’m willing to adjust my rating score if this claim is better supported.

2. Could the authors explain how Fig.4 is generated? Aren’t the initial several steps during the denoising process full of Gaussian noise? What does “projecting the current state back to the data manifold” mean (in Line 363)?

3. Could the authors elaborate more regarding the parsing function $\Psi$? How is such a parsing function defined and trained? Is it involved in the training, or only for the final evaluation?

**Limitations:**

Yes.

**Strengths And Weaknesses:**

Strengths:
- DiffThinker achieved quite high accuracy in evaluation benchmarking datasets after being continually trained via the flow matching objective.

Weaknesses:
- The paper claims that the denoising trajectory naturally serves as a visual reasoning trace without sufficient support (the only support is one qualitative example from a single task in Fig.4). Therefore, it remains unclear whether the current image editing models do have "native" reasoning capability and such reasoning capability could really be the underlying reason for the significant improvements in the models’ performances.
After the continual training of the image-editing model on individual tasks, could the improvements be due to the memorization of the diffusion model, given that the authors also admit that “out-of-distribution generalization remains constrained by the limited zero-shot reasoning proficiency of current generative foundation models” (in Line 862).

---

> ### Author Rebuttal · Authors · 2026-03-31
>
> >**W1 & Q1: More evidence is needed to support the claim "the denoising trajectory naturally serves as a visual reasoning path".
> Could the improvements be due to the memorization?**
>
> Thanks for your concern.
>
> 1.We want to clarify that DiffThinker’s performance comes from reasoning rather than simple memorization. We further reduce the intermediate steps and observe the model's performance. **If the model were merely memorizing and directly following a memorized correct path from blurry to sharp during denoising, the performance should not show a significant drop.** The results on VSP-Super level-32:
>
> | Steps | 20 | 10 | 5 | 4 | 3 | 2 |
> | :--- | :---: | :---: | :---: | :---: | :---: | :---: |
> | Accuracy (%) | 83 | 78 | 63 | 58 | 43 | 19 |
>
> At low step counts (e.g., 5 steps), generated paths remain visually clear but **exhibit logical errors like disconnections or unpruned forks**, causing a performance drop. This requirement for sufficient steps demonstrates that DiffThinker performs reasoning rather than simple input-output memorization.
>
>
>
> 2.During both intermediate steps and failure cases, DiffThinker generates results containing multiple paths, a state that **never exists in training**.
>
> 3.Following work on generative reasoning has **reached the same conclusion as ours**. In VBVR-Wan2.2 [1,2], which employs a training methodology **identical** to that of DiffThinker and DiffThinker-Video (Appendix A.4), the authors state in abstract [2]
> >We show that reasoning ... **emerges along the diffusion denoising steps**. Through qualitative analysis and targeted probing experiments, we find that **models explore multiple candidate solutions in early denoising steps and progressively converge to a final answer**.
>
> [1] A Very Big Video Reasoning Suite, arXiv:2602.20159
>
> [2] Demystifying Video Reasoning, arxiv:2603.16870
>
> 4.We further conduct experiments to elucidate how Classifier-Free Guidance (CFG) reflects reasoning intensity, providing evidence that DiffThinker performs reasoning.
>
> CFG represents the degree of constraint adherence in diffusion models. We introduce Exploration Intensity (EI) (Step 1 parsed path grids) to quantify this. We posit that EI distinguishes between reasoning and memorization: whereas memorization would produce a deterministic, memorized trajectory, reasoning involves a constraint-driven exploration to filter out invalid branches. We explore the variation of CFG from 1 to 4 to study the relationship between CFG and reasoning intensity while maintaining visual quality to ensure parsing. The results on VSP-Super level-32:
>
> | CFG | Acc | EI |
> | :--- | :---: | :---: |
> | 1 | 78 | 50.62 |
> | 2 | 80 | 63.70 |
> | 3 | 81 | 71.55 |
> | 4 | 83 | 77.93 |
>
> As CFG increases, the EI and accuracy concurrently improves. This indicates that low CFG scales diminish constraint adherence by suppressing the model's exploration, weakening reasoning and degrading performance. We contend that DiffThinker fundamentally performs reasoning rather than memorization.
>
>
> 5.In Jigsaw and VisPuzzle, every image is unique. In other tasks such as VSP-Super, Maze, TSP, and Sudoku, the combinatorial search space is extremely large. These make simple memorization practically impossible.
>
>
>
> ---
> >**Q2: Fig.4.**
>
> The intermediate latent $x\_t$ is indeed noisy, Figure 4 visualizes the model’s **estimated clean solution** $\hat{x}\_0$ at each step, rather than $x\_t$. Following Eq. 5-7, we obtain this estimate by "projecting back to the data manifold", using the predicted velocity $v\_\theta$ to map the current state $x\_t$ to estimated clean solution $\hat{x}\_0$: $\hat{x}\_0 \approx x\_t + (1-t) v\_\theta$, and then decode it into pixel space.
>
> ---
> >**Q3: Parsing function.**
>
> The parsing function, used only for final evaluation, first retrieves meta-information (such as grid size), then:
>
> VSP, VSP-Super, Maze:
> It partitions the image into grid cells, detects the distribution of red pixels (path signals), maps them to grid coordinates. Starting from the initial position, the visual trajectory is then reconstructed into an string until the trajectory terminates.
>
> TSP:
> It locates the centroids of yellow and blue nodes, mapping them to grid coordinates, then attempts to generate potential connections between any two points and calculates the pixel overlap ratio between these theoretical lines and the actual red path lines. If the number of parsed connections is valid, the process concludes by sequentially outputting the complete coordinate sequence starting from the origin.
>
> Sudoku, Jigsaw, VisPuzzle:
> It calculates the coordinate points for each grid cell. For Jigsaw, DiffThinker is trained to output a number at a fixed position in the top-left corner of each cell, enabling parsing based on these fixed coordinates. Each cell is individually processed using an OCR model (e.g., Qwen3-VL, not finetuned) to recognize the digits. The sequence of numbers is reconstructed into a string.
>
> We will add these details in appendix.

---

> > ### Author Rebuttal · Reviewer_Cpja · 2026-04-03
> >
> > Thanks for the reply. All my concerns have been resolved. I'll raise my rating score.

---

### Official Review · Reviewer_bqPu · 2026-03-13

**Soundness:** 3
**Presentation:** 3
**Significance:** 4
**Originality:** 2
**Overall Recommendation:** 4
**Confidence:** 4

**Summary:**

The manuscript introduces DiffThinker, a novel framework that establishes "Generative Multimodal Reasoning" by reformulating complex vision-centric tasks as image-to-image generation problems rather than standard text-centric symbolic mappings. Based on Qwen-Image-Edit and LongCat-Image-Edit, this paper interprets the iterative denoising trajectory as a continuous visual reasoning path. The authors evaluate this approach across diverse spatial and logical tasks, demonstrating that DiffThinker significantly outperforms proprietary models like GPT-5 and Gemini-3-Flash, as well as open-source baselines fine-tuned with both supervised learning and GRPO.

**Compliance With Llm Reviewing Policy:**

Affirmed.

**Final Justification:**

Thank you for the thorough rebuttal. Although the additional baseline comparisons are appreciated, the arguments for generalizability still rely more on cited future potential than direct empirical evidence. While I maintain a positive view of the paper's novelty, I will keep my original rating.

**Key Questions For Authors:**

See weaknesses.

**Limitations:**

Yes

**Strengths And Weaknesses:**

Strengths

1. The parallel interpretation and visualization is highly appreciated. Unlike standard autoregressive models that sequentially backtracks to correct errors or latent visual reasoning methods that struggle with effective parallelism, DiffThinker simultaneously explores multiple candidate trajectories in its early generative steps, progressively pruning invalid routes to converge on a globally consistent solution. Figure4 provides us with a clear and exciting illustration.

2. Efficiency and controllability are also good merits. When confronted with these vision-centric problem, text-space reasoning by MLLM is computationally expensive, yet ineffective. The framework provides a highly stable and controllable inference budget by relying on a fixed-step generative process, which is indeed efficient and effective.

3. The manuscript exhibits exceptional structural clarity and a compelling narrative logic. The authors concisely define and unpack four intrinsic properties of their proposed paradigm and the experimental section is then accordingly organized. This tightly coupled progression between theoretical claims and empirical validation significantly enhances the paper's readability and the overall claim of the framework.

Weaknesses

1. While DiffThinker demonstrates impressive performance on its evaluated benchmarks, its reliance on task-specific training paradigms inherently limits its broader applicability. The methodology involves training independent, specialized models for VSP, Maze, TSP, Sudoku, and Jigsaw. This siloed approach restricts the model's overall generalizability and its out-of-distribution zero-shot reasoning capabilities. By contrast, advanced generative models Nano Banana could perform in a zero-shot manner across various scenarios.

2. The current evaluation suite is heavily skewed toward fixed, deterministic, and highly structured patterns (e.g., grid-based environments and coordinate-bound TSP). Although the inclusion of VisPuzzle and Jigsaw  adds some variety, these still represent relatively constrained spatial problems. A critical unanswered question is whether the diffusion model's generative reasoning and spatial precision will severely degrade when faced with highly complex, unstructured visual inputs. I strongly recommend evaluating the model on more generalized benchmarks that demand robust visual imagination, such as BabyVision, to rigorously test the boundaries and scalability of this generative paradigm.

3. Fundamentally, DiffThinker operates on a "Thinking with Generative Models" philosophy. Including a direct comparison with a state-of-the-art baseline like Nano Banana would provide a much clearer picture of DiffThinker's relative advantages and fully substantiate its claims of superiority in generative multimodal reasoning.

Overall, this is a very strong, thought-provoking submission that pushes the boundaries of multimodal reasoning. The proposed paradigm is highly innovative and well-executed. If the authors can address these empirical gaps during the rebuttal, particularly by providing the suggested baseline comparisons **or** testing on a more complex, generalized benchmark like BabyVision, I would pose a much better estimation of this paper.

---

> ### Author Rebuttal · Authors · 2026-03-31
>
> >**W1: Task-specific training, generalizability and out-of-distribution zero-shot reasoning capabilities**
>
> Regarding task-specific training: As the first exploration of the Generative Multimodal Reasoning paradigm, our **primary objective is to demonstrate the intrinsic potential of this approach**, for instance, by showing it can outperform larger MLLMs like Qwen3-VL-32B under consistent training conditions, and by revealing **key properties** of generative reasoning such as native parallelism. This training approach facilitates our rigorous quantitative study of these fundamental properties.
>
> Also, as explored in **Appendix A.3**, our preliminary experiments with joint training demonstrate that the model can maintain competitive performance across domains, providing an initial proof-of-concept for its scalability into a unified system.
>
> While current resource constraints limit our ability to conduct the laege-scale training required to fully verify broad generalization, we believe that **Generative Multimodal Reasoning possesses the potential for out-of-distribution zero-shot reasoning.** Following work on generative reasoning indicates that training with large-scale and diverse data can further stimulate reasoning capabilities. For example, VBVR [1] constructed millions of data samples across hundreds of synthetic tasks and employed **the same method** as DiffThinker and DiffThinker-Video to train diffusion models for reasoning, demonstrating out-of-distribution zero-shot reasoning capabilities across 50 tasks. This suggests that given **sufficient resources and large-scale data**, DiffThinker has the clear potential to generalize to diverse tasks and scale into an all-purpose reasoning model.
>
> [1] A Very Big Video Reasoning Suite, arXiv:2602.20159
>
> ---
> >**W2: The reviewer recommend evaluating on more generalized benchmarks such as BabyVision.**
>
> As stated in W1, we primarily focus on **demonstrating the intrinsic properties** of the Generative Multimodal Reasoning paradigm. Due to current limitations in computational resources and large-scale training data, it is challenging to generalize to BabyVision at this stage. However, we believe that given sufficient resources and large-scale data, DiffThinker and DiffThinke-Video has the clear potential to scale to handle unstructured tasks, as shown in following work [1].
>
> ---
> >**W3: Including a direct comparison with a state-of-the-art baseline like Nano Banana.**
>
> Following the reviewer's valuable advice, we have conducted a direct comparison with a state-of-the-art baseline. We use the latest Nano Banana2.
>
> Table 1: Results on VSP, VSP-Super and Maze
> | Model | VSP 3 | 4| 5 | 6 | 7 | 8 | VSPS 16 | 32 | Maze 8 | 16 | 32 |
> | :--- | :---: | :---: | :---: | :---: | :---: | :---: | :---: | :---: | :---: | :---: | :---: |
> | Qwen-Image-Edit-2509 | 33 | 36 | 22 | 12 | 11 | 7 | 0 | 0 | 0 | 0 | 0 |
> | **DiffThinker** | **99** | **100** | **98** | **99** | **100** | **100** | **96** | **83** | **100** | **97** | **56** |
> | Nano Banana 2 |75 |66 |41 |42 |35 |35 |6 |7 |1 |0 |0 |
>
>
> Table 2: Results on TSP, Sudoku, Jigsaw, Vispuzzle
> | Model | TSP 12 | 15 | 18 | Sudoku 45 | 40 | 35 | Jigsaw 2 | 3| 4| VisP. |
> | :--- | :---: | :---: | :---: | :---: | :---: | :---: | :---: | :---: | :---: | :---: |
> | Qwen-Image-Edit-2509 | 0 | 0 | 0 | 0 | 0 | 0 | 0 | 0 | 0 | 7.5 |
> | **DiffThinker** | **74** | **62** | **58** | **98** | **95** | **57** | **99** | **97** | **80** | **98.3** |
> | Nano Banana 2 |0 |0 |0 |10 |5 |0 |11 |0 |0 |39 |
>
> While Nano Banana 2 demonstrates zero-shot capabilities even in complex tasks including VSP-Super, Sudoku, and Jigsaw, its overall performance remains limited. This performance gap highlights both the significant difficulty of these vision-centric reasoning tasks and the superior proficiency of DiffThinker. We will **update these results in the main text and provide additional visualizations for each task in the Appendix**.
>
> ---
> >**If the authors can provide suggested baseline (Nano Banana) **or** test on BabyVision, the reviewer would pose a much better estimation of this paper.**
>
>
>
> We sincerely thank the reviewer for the highly positive evaluation and for recognizing DiffThinker as a "very strong, thought-provoking" and "highly innovative" contribution. To address the empirical gaps mentioned:
>
> Regarding baseline comparisons, we have conducted a direct comparison with Nano Banana2.
>
> Regarding more complex benchmarks like BabyVision, while current resource constraints prevent us from conducting the large-scale training required for such broad generalization at this stage, we have provided evidence from following work (VBVR) and we believe that Generative Multimodal Reasoning can achieve out-of-distribution zero-shot reasoning when **scaled with sufficient data and training resources**.

---

> > ### Author Rebuttal · Reviewer_bqPu · 2026-04-04
> >
> > I thank the authors for the thorough rebuttal. The Nano Banana 2 comparison (W3) is well-addressed and appreciated.
> >
> > However, my core concerns (W1, W2) remain partially unresolved. The argument for generalizability relies primarily on citing a follow-up work (VBVR) rather than direct empirical evidence within this submission. For a paper proposing a new paradigm, I believe stronger evidence of generalization beyond task-specific, structured settings is essential — and this would require substantial additional experimentation beyond a rebuttal cycle.
> >
> > I maintain my positive view of the paper novelty and look forward to broader evaluations in future versions.

---

### Official Review · Reviewer_tSKv · 2026-03-13

**Soundness:** 3
**Presentation:** 3
**Significance:** 3
**Originality:** 3
**Overall Recommendation:** 4
**Confidence:** 4

**Summary:**

The paper proposes DiffThinker, a diffusion-based framework for generative multimodal reasoning that performs reasoning directly in visual space rather than through textual chain-of-thought. It models reasoning as an image-to-image denoising process, where intermediate states represent reasoning steps. The approach is evaluated on seven tasks across planning, optimization, constraint satisfaction, and spatial reasoning, and shows stronger performance than several baseline MLLMs while demonstrating advantages in efficiency, controllability, and parallel reasoning.

**Compliance With Llm Reviewing Policy:**

Affirmed.

**Final Justification:**

In the authors' rebuttal, they provide detailed analysis to address my concerns. Thus, I maintain my positive scores.

**Key Questions For Authors:**

1. Generalization to real-world tasks:
 The current evaluation focuses on structured tasks such as Sudoku, TSP, and maze planning. How does DiffThinker perform on more realistic multimodal reasoning benchmarks such as visual question answering, embodied planning, or document understanding?
2. Fairness of baseline comparisons:
 DiffThinker is trained separately for each task using task-specific datasets. Were the baseline MLLMs also fine-tuned under similar conditions? If not, could the performance gap partly reflect differences in task-specific training?
3. Scalability of the approach:
 The paper mentions training separate models for each task. Could a single DiffThinker model generalize across multiple reasoning tasks, similar to general-purpose MLLMs?
4. Interpretability of diffusion reasoning trajectories:
 The authors interpret denoising steps as reasoning steps. Have the authors analyzed whether these intermediate states correspond to semantically meaningful reasoning operations?
5. Integration with language reasoning:
 How would the authors extend DiffThinker to tasks that require both spatial reasoning and symbolic reasoning simultaneously?

**Limitations:**

Yes.
The authors explicitly discuss potential societal impacts and limitations, including possible misuse of generative models for producing misleading visual information and the environmental costs associated with training large generative models.

**Strengths And Weaknesses:**

Strengths
1. Interesting paradigm shift for multimodal reasoning
The main contribution of the paper is conceptual. Instead of treating reasoning as text generation, the authors reformulate reasoning as a visual generative process using diffusion models. This shift from symbolic space to visual space is a novel and intriguing direction. The formulation of reasoning trajectories through the diffusion denoising process provides an intuitive interpretation of intermediate reasoning states.
2. Insightful discussion of reasoning properties
The authors provide a thoughtful conceptual discussion of the properties of generative multimodal reasoning, highlighting four attributes: efficiency, controllability, native parallelism, and collaboration with language models. This perspective contributes to a broader understanding of how reasoning may emerge in generative models.
3. Potential synergy with MLLMs
The collaborative pipeline between DiffThinker and MLLMs is an interesting idea. Using the diffusion model to generate candidate visual solutions and an MLLM to verify them suggests a hybrid reasoning architecture that could be promising in future systems.

Weaknesses
1. Benchmark tasks are relatively synthetic
Most experiments are conducted on structured tasks such as Sudoku, maze navigation, TSP, and jigsaw puzzles. While these tasks test spatial reasoning ability, they are relatively synthetic and may not reflect real-world multimodal reasoning scenarios. It remains unclear whether the approach would generalize to more complex real-world tasks such as robotics planning, document reasoning, or visual question answering.
2. Comparisons with MLLMs may not be entirely fair
The comparison between diffusion-based reasoning and autoregressive MLLMs is interesting but raises questions about fairness. MLLMs are not typically optimized for these specific structured tasks, while DiffThinker is trained separately on task-specific datasets. This training setup could partly explain the performance gap.
3. Limited evaluation on open-ended reasoning
The proposed paradigm is evaluated mainly on tasks with clear visual outputs and deterministic parsing rules. It is unclear how the framework would handle open-ended reasoning tasks where intermediate states are not easily represented as images.
4. Training cost and scalability are not fully analyzed
Although the paper discusses training and inference efficiency, the overall training cost of training separate models for multiple tasks may still be significant. A more detailed discussion of scalability and resource requirements would strengthen the paper.

---

> ### Author Rebuttal · Authors · 2026-03-31
>
> >**W1 & W3 & Q1: Application to real-world and open-ended tasks such as robotics or VQA.**
>
> We first explain that our current focus on synthetic environments is primarily intended for  quantitative study (**Line 249**). These structured tasks provide controllable and scalable difficulty levels, which facilitate a precise exploration of the model's reasoning capabilities and boundaries.
>
> Also, this paradigm is not limited to such scenarios. As demonstrated by recent research such as MMGR [1], when built upon powerful closed-source image or video generation models (consistent with our exploration of DiffThinker and DiffThinker-Video in Appendix A.4), Generative Multimodal Reasoning can be successfully applied to complex real-world tasks including 3D Real-World Navigation and Math VQA. Specifically, for 3D navigation, a video generation model can receive a 3D scene image and a navigation goal as input, directly generating a video that illustrates the robot's complete trajectory from the starting point to the destination. For Math VQA, an image generation model can take a math problem as input and produce a static image that visually writes out steps of solution process and final answer. These results show that Generative Multimodal Reasoning is a versatile framework capable of representing state evolution even in open-ended real-world environments.
>
> [1] MMGR:Multi-Modal Generative Reasoning, arXiv:2512.14691
>
>
> ---
> >**W2 & Q2: Comparison with MLLMs.**
>
>
> **We have trained both DiffThinker and the baseline open-source MLLMs (e.g., Qwen3-VL-32B) separately on identical task-specific datasets for fairness**. As detailed in **Appendix B.1.2 and Table 5**, the MLLM baselines were fine-tuned using **the same data** distribution，ensuring that the performance gap comes directly from the Generative Multimodal Reasoning paradigm itself, rather than differences in training data.
>
>
> >**W4 & Q3: Training cost for separate models and scalability to general-purpose model.**
>
>
> As discussed in W2, both DiffThinker and the MLLM baselines (e.g., Qwen3-VL-32B) were trained under identical data distributions, which ensure a fair comparison. Regarding the concern over training separate models: because we maintain a fixed total data volume and number of epochs, the cumulative computational cost of training specialized models for each individual task is **mathematically equivalent** to training a single multi-task model on the mixed dataset.
>
> Also, as the first exploration of the Generative Multimodal Reasoning paradigm, our **primary objective is to demonstrate the intrinsic potential and properties** of this approach. We believe a single DiffThinker model **can generalize across multiple reasoning tasks**，with large-scale training. Following work on generative reasoning indicates that training with large-scale and diverse data can further stimulate reasoning capabilities. For example, VBVR [1] constructed millions of data samples across hundreds of synthetic tasks and employed **the same method** as DiffThinker and DiffThinker-Video to train diffusion models for reasoning, demonstrating strong generalization abilities across 50 tasks. This suggests that given sufficient resources and large-scale data, DiffThinker has the clear potential to generalize and scale into an all-purpose reasoning model.
>
> [1] A Very Big Video Reasoning Suite, arXiv:2602.20159
>
> ---
> >**Q4: Interpretability of diffusion reasoning trajectories.**
>
> Actually, we have analyzed these intermediate states and show that **they correspond to semantically meaningful reasoning operations**, functioning as a native parallel reasoning path. As illustrated in **Figure 4** and discussed in **Section 4.2** under "DiffThinker as a Native Parallel Reasoner," the denoising trajectory provides a view of the reasoning process where early steps (e.g., Steps 1 and 4) clearly correspond to parallel exploration, allowing the model to investigate multiple candidate trajectories across the grid to avoid premature commitment. Subsequent stages, such as Step 7, then correspond to the pruning of invalid routes and the refinement of the optimal path based on global constraints. This visual progression demonstrates that the iterative denoising process is an interpretable sequence of logical state transitions that effectively tracks the evolution of reasoning directly within the visual space.
>
> ---
> >**Q5: Integration with language reasoning.**
>
>
> **We have discussed in Section 4.2**, “DiffThinker as a Collaborative Partner” and proposed integrating DiffThinker with language models. This synergy is demonstrated on Jigsaw level-4, which demands both precise spatial configuration and logical verification. In this collaborative framework, DiffThinker first leverages its visual imagination to generate multiple candidate solutions, then an MLLM evaluates these candidates to make a final decision. As shown in Figure 6, this partnership achieves superior accuracy.

---

### Official Review · Reviewer_obaq · 2026-03-17

**Soundness:** 3
**Presentation:** 3
**Significance:** 3
**Originality:** 3
**Overall Recommendation:** 4
**Confidence:** 4

**Summary:**

The papper proposes a method in Image to Image generation as the generative thinking paradigm that directly reasons in the diffusion process. It novelly transform the visual reasoning in the generative I2I mode and skips the symbolic reasoning that maps multimodal input into textual space. DiffThinker proformes in the native visual space and can track the reasoning transitions directly in the visual space and has the potential in long-horizon, visual centric tasks. By learning to directly "draw" the solutions (like drawing a line through a maze), the model can track visual changes over time. The paper addresses an important concept by evaluating this method on spatial and structured tasks, such as navigating mazes, finding the shortest routes (TSP), solving Sudoku, and putting together jigsaw puzzles.


It proposes four core characteristics, efficiency, controllability, parallelisam, and collaboration. On 7 benchmarks, DiffThink has outperformed SOTA MLLMs, including proprietary models.

**Compliance With Llm Reviewing Policy:**

Affirmed.

**Final Justification:**

The reviewer is satisfied with the rebuttals and reinforced my prior assessment.

**Key Questions For Authors:**

1. DiffThinker as collaborative partner is inferences at step upperbound of 3 to 5. The gain from MLLMs look weak. Is there any gain on step 20 as stated in 4.3?
2. DiffThinker is a great pure visual planning machanism, however, also restricted to solving only task with clear initial state and end state. The reviewer believe this limitation is inherited from flow matching and doubt whether it would be a general visual reasoning framework in the future.

**Limitations:**

Yes.

**Strengths And Weaknesses:**

Major strengths:
1. Moving reasoning from text tokens to visual space is a smart idea. Using the image generation process as a "visual Chain-of-Thought" helps avoid the common struggles that text-based models face when trying to track 2D spatial relationships over many steps.
2. The method achieves great accuracy gains over leading industry models and the fine-tuned Qwen3-VL baseline. The paper also includes good extra tests, like comparing their image-based approach to a video generation model (DiffThinker-Video).
3. The authors analyzed how their method works comprehensively. They clearly explain the effect of different generation steps, how the model explores multiple visual paths at the same time (native parallel reasoning), and how it can team up with text-based models to get even better results.

Major weakness:
1. The tested datasets are mostly specialized spatial puzzles rather than the "general multimodal reasoning" described in the paper. As the authors state, the model struggles to generalize to unseen tasks outside of its training data. The visual reasoning tasks are strongly limited to tasks with a deterministic final state instead of optimizing the intermediate steps on the fly. Does DiffThinker limited to visual tasks that are dynamic dependent to the action performed in each task?
2. The method needs a separate parsing tool to translate the generated image back into text to check if the answer is right. It is unclear how well this parser handles small visual errors or messy drawings. For example, is there intersective paths in the maze that will lead to ambiguous step order that is hard to parse? Has the mapping error from generated path to symbolic representations been calculated and how do they affect the final plan? Instead of a hard-coded parser, using another Vision-Language model to look at the final image and grade it might be a more natural and useful approach.
3. The authors had to train separate, independent models for each type of puzzle. When they tried mixing the tasks together (joint training), the model's performance actually dropped on the harder tasks (e.g., Maze level-32 accuracy dropped from 56% to 50%). This makes it hard to scale the method into a single, all-purpose reasoning model.
4. Regarding latency. In 4.2 the authors claim that DiffThinker has competitive latency of 1.1s instead of 1.4s for QwenVL-32B 1.4s. However, in 4.3 the authors mentioned that they use the best inference time of 20 steps as the default configurations. According to Figure 3  the inference time increases a lot when the steps become 30. The reviewer doubt that with more complex visual planning tasks, the latency claim will become weaker and even reversed.
5. Parallelism of DiffThinker is not compared with MLLMs with test-time scaling, which also ensures great parallel reasoning.

Minor weaknesses
1. The paper does not clearly explain how the visual model learns the strict math and logic rules of Sudoku just by looking at pictures of grids.
2. The comparision on Gemini-3-Flash is somewhat questionable as the other proprietary model is GPT-5, while Gemin Flash model is much weaker than Gemini-3 or Gemini-3 Pro. Considering the outstanding results in VSP regarding Gemini-3-Flash, a test results on Gemini-3 or 3 pro is a more fair baseline.

---

> ### Author Rebuttal · Authors · 2026-03-31
>
> >**W1 & Q2: General visual reasoning, limitation to tasks with clear initial and end state.**
>
> First, we clarify a potential misunderstanding: we propose "**Generative** Multimodal Reasoning" rather than "General Multimodal Reasoning".
>
> Also, general multimodal reasoning tasks can be fundamentally organized into an Input-Reasoning-Output framework. Within this structure, the problem input naturally serves as the starting state, while the solution represents the end state. Thus, Diffthinker is not limited to task with clear initial and end states.
>
> ---
> >**W2: How the parsing tool handles errors, how scores are calculated, why not use VLM.**
>
> In cases where the model produces messy drawings or ambiguous branches, the tool will identify these as parsing errors. An example is provided in **Appendix, Figure 18**.
>
> Any parsing error is counted as a failure of DiffThinker. Such rendering errors become exceptionally rare after training.
>
> The primary reason for not using a VLM is that current VLMs exhibit lower performance than DiffThinker in these tasks and **may introduce false positives** by misjudging incorrect answers as correct, which would be unfair to MLLM baselines. As discussed in **Line 203**, our tool-based parsing ensures fairness.
>
> ---
> >**W3: Scale the method into an all-purpose reasoning model.**
>
> As shown in Appendix A.3 Table 3, joint training causes no significant performance drop; in fact, accuracy on some tasks such as VSP-Super level-16 even improves from 96 to 100.
>
> Moreover, as the first exploration of the Generative Multimodal Reasoning paradigm, our **primary objective is to demonstrate the intrinsic potential and unique properties** of this approach. Following work on generative reasoning indicates that training with large-scale and diverse data can further stimulate reasoning capabilities. For example, VBVR [1] constructed millions of data samples across hundreds of synthetic tasks and employed **the same method**（Flow Matching）as DiffThinker and DiffThinker-Video to train diffusion models for reasoning, demonstrating generalization abilities across 50 tasks. This suggests that given sufficient resources and large-scale data, DiffThinker has the clear potential to **scale into an all-purpose reasoning model**.
> [1] A Very Big Video Reasoning Suite, arXiv:2602.20159
>
> ---
> >**W4: Latency in complex problems.**
>
> This is not a weakness but an advantage of our paradigm: controllability. When facing complex problems, MLLMs generate significantly longer CoT, leading to unpredictable and prohibitive computational costs. In contrast, DiffThinker ensures a deterministic computational budget by **allowing users to fix the number of inference steps, regardless of task**. As shown in **Figure 7**, even for our most difficult task Maze level-32, 20 steps is sufficient to achieve good performance. Thus, the latency advantage will not reverse in complex tasks.
>
> ---
> >**W5:Test-time scaling (TTS) of MLLM.**
>
> We clarify a potential misunderstanding: the parallelism of DiffThinker is native, not an external TTS strategy, and will **not increase computational complexity**. While MLLMs can simulate parallel reasoning through TTS methods (e.g., Best-of-N), these are external strategies that incur significant computational overhead. In this work, we primarily compare the **intrinsic capabilities** of each paradigm (Line 358) under comparable computational budgets, as demonstrated in Figure 5.
>
> ---
> >**W6: How the visual model solves Sudoku?**
>
> DiffThinker is built upon Qwen-Image-Edit, which uses Qwen2.5-VL as Encoder, which already possesses an inherent understanding of rules of Sudoku.
>
> Moreover, DiffThinker treats reasoning as a conditional generation task based on the provided numerical clues. The iterative denoising trajectory naturally serves as the reasoning process. As the model progressively refines the image, it integrates global constraints to ensure the output is logically consistent with the rules of Sudoku.
>
> ---
> >**W7: Gemini-3-Flash is much weaker?**
>
> Gemini-3-Flash is not much weaker in multimodal reasoning; it frequently **matches or even surpasses** Gemini-3 Pro. For instance, on MMMU-Pro, Gemini-3-Flash scores 81.2%, higher than Gemini-3 Pro’s 81.0% (source: Google’s official technical documentation). We are currently unable to test Gemini-3-Pro because its API has been deprecated by Google.
>
> ---
> >**Q1: DiffThinker as a collaborative partner.**
>
>
> There appears to be a misunderstanding. The values **3 and 5 refer to the number of candidates (images generated by DiffThinker)**, analogous to Best-of-N, rather than diffusion steps. Each of these is generated using the default **20 steps**. The MLLM then selects the final solution from these .
>
> **The gain of MLLM is not weak.** The MLLM alone scores 0 on Jigsaw level-4, and DiffThinker alone scores 80. By collaborating ($N=3$), the system performance improves to 84. The results for $N=5$ further show performance gains over the $N=3$ configuration.

---

> > ### Author Rebuttal · Reviewer_obaq · 2026-04-04
> >
> > The reviewer acknowledges and thanks the clarifications of the authors for their clarifications and is clearer with the scope and effectiveness of the model.

---

### Decision · Program_Chairs · 2026-04-30

**Decision:**

Accept (regular)

**Comment:**

This paper introduces DiffThinker, a diffusion-based reasoning framework that reframes multimodal reasoning as a native image-to-image generative task, rather than the text-centric chain-of-thought approach dominant in MLLMs. The rebuttal efficiently addresses the reviewers’ concerns.

While we agree with the reviewers that stronger evidence of generalization beyond task-specific, structured settings is essential, the authors’ arguments for generalizability still rely more on cited future potential than on direct empirical evidence. Therefore, we decide to accept this paper, but we urge the authors to include additional empirical evidence in the camera-ready version.